

# Secondary organic aerosol production from pinanediol, a semi-volatile surrogate for first-generation oxidation products of monoterpenes

Penglin Ye[a], Yunliang Zhao, Wayne K. Chuang, Allen L. Robinson, Neil M. Donahue[*]

Center for Atmospheric Particle Studies, Carnegie Mellon University, 5000 Forbes Avenue, Pittsburgh, Pennsylvania 15213, United States

[a]now at: Aerodyne Research Inc, Billerica, MA 01821, USA

*Correspondence to: nmd@andrew.cmu.edu

Phone: (412) 268-4415





**Abstract**
We have investigated the production of secondary organic aerosol (SOA) from pinanediol
(PD), a precursor chosen as a semi-volatile surrogate for first-generation oxidation
products of monoterpenes. Observations at the CLOUD facility at CERN have shown that
oxidation of organic compounds such as PD can be an important contributor to new-particle
formation. Here we focus on SOA mass yields and chemical composition from PD photo-
oxidation in the CMU smog chamber. To determine the SOA mass yields from this semi-
volatile precursor, we had to address partitioning of both the PD and its oxidation products
to the chamber walls. After correcting for these losses, we found OA loading dependent
SOA mass yields from PD oxidation that ranged between 0.1 and 0.9 for SOA
concentrations between 0.02 and 20 µg m$^{-3}$, these mass yields are 2–3 times larger than
typical of much more volatile monoterpenes. The average carbon oxidation state measured
with an Aerosol Mass Spectrometer was around -0.7. We modeled the chamber data using
a dynamical two-dimensional volatility basis set and found that a significant fraction of the
SOA comprises low volatility organic compounds that could drive new-particle formation
and growth, which is consistent with the CLOUD observations.



## 1 Introduction

Particulate matter (PM) in the atmosphere affects human health and life expectancy (Pope et al., 2009) and also influences Earth's climate by absorbing and scattering radiation (Solomon, 2007). Organic compounds constitute a large fraction of that PM, making up around 20–90% of the aerosol mass in the lower troposphere (Kanakidou et al., 2005). Secondary organic aerosol (SOA), formed from oxidation of gas-phase organic compounds in the atmosphere, accounts for a significant fraction of the organic aerosol (OA) in PM (Zhang et al., 2007). In the atmosphere, OA is dynamic due to constant photo-oxidation and associated evolution in thermodynamic properties (Seinfeld and Pandis, 2006; Donahue et al., 2005). However, classical smog-chamber experiments encompass only the early stages of SOA formation, including one generation or at most a few generations of oxidation chemistry (Pandis et al., 1991; Odum et al., 1996a). While those experiments may include some later-generation chemistry, the commonly used two-product model (Odum et al., 1996a) treats the (quasi) first-generation products as effectively non-reactive.

Further oxidation (aging) of SOA may add more functional groups to the carbon backbone, causing the second-generation oxidation products to be even less volatile and more water soluble than the first-generation products, which will also enhance the SOA mass (Donahue et al., 2005). However, ongoing oxidation must eventually fragment products and drive down the SOA mass because the end state of organic oxidation is $CO_2$ formation (Kroll et al., 2009; Chacon-Madrid et al., 2012; Donahue et al., 2013). There is considerable evidence that the ongoing oxidation chemistry can increase SOA mass and oxidation state, both from smog-chamber experiments (Donahue et al., 2012a; Henry and Donahue, 2012; Qi et al., 2012) and also from flow tubes that simulate many days of oxidation using intense



UV radiation to drive photochemistry (Lambe et al., 2011; Wong et al., 2011; Cubison et
al., 2011). The flow-tube results also confirm that oxidation will eventually cause mass
loss via fragmentation (Tkacik et al., 2014). The volatility basis set (VBS) was developed
to treat this ongoing chemistry by condensing the enormous ensemble of organic
compounds involved onto a basis grid described by volatility and the carbon oxidation state
(Donahue et al., 2006; Donahue et al., 2011a; Donahue et al., 2012b; Chuang and Donahue,
2016b; Tröstl et al., 2016), with coupling constants constrained by chemical behavior of
representative or average compounds (Chacon-Madrid et al., 2012; Donahue et al., 2013).
Bulk SOA aging experiments show that later-generation chemistry will influence SOA
properties, but those experiments provide limited mechanistic insight due to the extreme
complexity of the chemistry involving multiple generations of multiple products. A
complementary approach is to use selected first-generation products from SOA formation
to probe    second-generation    chemistry    systematically,    and    to    proceed    through
representative later-generation products. For example, the known products of α-pinene
oxidation include pinonaldehyde, which is one of the most volatile products, and acids such
as cis-pinonic acid and pinic acid which are some of the least volatile monomer products
(Jang and Kamens, 1999; Jaoui and Kamens, 2001). Smog-chamber experiments at
Carnegie Mellon have shown that pinonaldehyde is a modest but significant source of SOA
at both high NO (Chacon-Madrid and Donahue, 2011) and low NO (Chacon-Madrid et al.,
2013) conditions. Aldehyde chemistry is dominated by OH radical attack on the terminal -
CHO moiety, causing fragmentation (Chacon-Madrid et al., 2010), but OH attack along
the carbon backbone leads to functionalized products that condense to enhance SOA
formation from the first-generation parent α-pinene, with mass yields of roughly 10%



under atmospherically relevant conditions. If the most volatile α-pinene product can
enhance SOA production, it stands to reason that less volatile SVOC products would have
an even greater effect. Indeed, we have observed very low volatility products from cis-
pinonic acid oxidation, such as MBTCA (Müller et al., 2012), but we have not
systematically explored the SOA mass yields from first-generation SVOC products. Here
we use pinanediol (PD) as a surrogate for semi-volatile first-generation oxidation products
of monoterpenes to study this aging chemistry. PD has a volatility similar to cis-pinonic
acid ($C^* \sim 300\ \mu g\ m^{-3}$) but it is commercially available and easier to handle.
One reason that SOA mass yields from SVOCs are not commonly reported is that SVOCs
are hard to handle and measure, and mass-yield determinations require accurate values for
the amount of oxidized precursor because the mass yield by definition is the ratio of formed
SOA to oxidized precursor mass. There are two reasons why this is challenging for SVOCs.
First, they are sticky and hard to measure. Second, and more challenging, SVOCs may be
lost to Teflon chamber walls (Matsunaga and Ziemann ‡, 2010) and may even return from
the chamber walls as oxidation perturbs a putative gas-Teflon equilibrium. This means any
measured change in the SVOC concentration, even if an instrument is well characterized,
may not reflect the actual amount of oxidized SVOC.
Sorption of SVOCs into Teflon chamber walls has recently become a matter of significant
concern. Matsunaga and Ziemann (2010) showed that various organic compounds broadly
in the intermediate volatility range (IVOCs, (Donahue et al., 2011a)) appear to sorb
reversibly to Teflon chamber walls, and more recent work has confirmed this finding. The
fraction of organic vapors left in gas phase appears to depend on the volatility and the
molecular structure of the organics, but Matsunaga and Ziemann suggested that IVOCs



partition into a disrupted surface layer of the Teflon as if the Teflon had an equivalent mass
of between 2 and 10 mg m$^{-3}$, depending on molecular structure (for a several cubic meters
chamber). As an example, an 8 m$^3$ chamber has a surface area of 12 m$^2$, and if the disrupted
Teflon surface layer postulated by Matsunaga and Ziemann were 1 µm thick it would have
a volume of 12 x 10$^{-6}$ m$^3$ and thus a mass of roughly 10 g; projected to the chamber volume
this gives an equivalent mass concentration of roughly 1 g m$^{-3}$. To have an effective
"partitioning mass" of 1-10 mg m$^{-3}$ this material would thus need to have a mass-based
activity coefficient of 100-1000 (Trump et al., 2016). This is consistent with weak
interactions involving non-polarizable Teflon and also a low degree of interactions among
sorbed organics within the walls at the Henry's law, low-concentration limit. However, we
must stress that the exact mechanism of organic sorption to Teflon chamber walls remains
unclear.
More recently, Ye et al. (Ye et al., 2016a) and Krechmer et al. (Krechmer et al., 2016)
showed that SVOCs are lost to the Teflon walls steadily, with a time constant of roughly
15 minutes (again for a several cubic-meter chamber). The SVOCs in these studies had 1
$< C* < 300$ µg m$^{-3}$ and so would be expected to leave only a small fraction ($\ll 10\%$) in the
gas phase; this quasi-irreversible loss is thus broadly consistent with the reversible
equilibration reported earlier for IVOCs.
We expect PD to partition substantially to the walls of a Teflon chamber. Even 2-decanol
showed significant vapor loss (Matsunaga and Ziemann, 2010), and the additional OH
group in PD decreases the vapor saturation concentration of PD by around 2.3 decades
(Donahue et al., 2011a). This should cause larger mass loss to the chamber walls. In order
to get an accurate SOA mass yield from oxidation of PD, we need to determine how much



PD exists in the gas phase vs the chamber walls, and ultimately how much PD reacts during
SOA formation experiments.
Another reason we are interested in SOA formation from PD is that it has already been
used as a surrogate for the first-generation terpene oxidation products to explore the role
of gas-phase aging in new-particle formation, and we wish to compare SOA formation with
new-particle formation. The Cosmics Leaving OUtdoor Droplets (CLOUD) facility at
CERN is designed to study the effects of cosmic rays on new-particle formation (nucleation
and growth) (Kirkby et al., 2011; Duplissy et al., 2016). Early experiments focused on
sulfuric acid vapor and different stabilizing species that include the ammonia, amines and
oxidation products of organic precursors (Kirkby et al., 2011; Schobesberger et al., 2013;
Riccobono et al., 2014). PD was used to mimic first-generation oxidation products of
monoterpene formed in the atmosphere (Schobesberger et al., 2013). Specifically the
experiments addressed the hypothesis that oxidation of these first-generation products by
OH radicals could produce later-generation products with sufficient supersaturation to
participate in nucleation (Donahue et al., 2011c). The PD oxidation experiments were
among the first to observe highly oxidized, extremely low volatility organic compounds
(ELVOCs) (Donahue et al., 2011a), with the original 10 carbon atoms decorated by up to
12 oxygen atoms (Schobesberger et al., 2013; Riccobono et al., 2014). The composition of
these highly oxidized organic molecules (HOMs) and possible mechanisms for their
formation remains an active research topic (Ehn et al., 2014).
In this study, we focus on SOA formation following oxidation of PD by OH radicals. Our
first objective is to extend our understanding of SOA aging via experiments addressing
carefully selected first-generation products from common SOA precursors. Our second



objective is to compare the properties of bulk SOA produced at relatively high
concentrations (0.3-30 µg m$^{-3}$) with the PD oxidation products observed condensing onto
particles during the CLOUD nucleation experiment. Our third objective is to use PD as a
model compound to explore the complications of precursor losses to Teflon walls in smog-
chamber SOA formation experiments. We explore the wall sorption of PD by comparing
the total amount of PD injected into the chamber to the PD concentration observed in the
gas phase. We also investigate the release of sorbed PD from the chamber walls by heating
or diluting the chamber. We then calculate the SOA mass yields, accounting for the loss of
PD and also the loss of oxidation products to the Teflon chamber walls. Finally, we
describe the elemental composition of the formed SOA. We analyze the SOA volatility
distribution and oxidation state within the two-dimensional volatility-oxidation set (2D-
VBS) and compare the properties of bulk SOA to the ELVOCs observed in CLOUD.
**2 Materials and methods**
We conducted experiments in the Carnegie Mellon University (CMU) Smog Chamber, a
10 m$^{-3}$ Teflon bag suspended in a temperature-controlled room. The chamber and our
methodology have been described extensively in the literature (Hildebrandt et al., 2009).
Before each experiment, we cleaned the bag by flushing it with clean, dry air and exposing
it to UV irradiation at ~35 °C. We subsequently maintained the chamber at a constant
temperature unless otherwise noted.
For the experiments in this paper, we introduced organic compounds into the chamber via
a flash vaporizer (Robinson et al., 2013). We used a small, resistive metal heater enclosed
in a stainless-steel sheath to evaporate the organics inside the chamber, placing the organics



into an indentation on the stainless-steel surface before inserting the heater into the
chamber on the end of a long stainless-steel tube. With a flow of clean, dry dispersion air
flowing through the tube for mixing, we power-cycled the heater until the organics
completely evaporated. For various experiments, we used *n*-tridecane, 1-tridecene, 2-
nonanone, 2-nonanol, oxy pinocamphone, and pinanediol (Sigma-Aldrich, 99%). For SOA
formation experiments we used ammonium sulfate seed particles ($(NH_4)_2SO_4$, Sigma
Aldrich, 99.99%), which we formed by atomizing a 1 g $L^{-1}$ $(NH_4)_2SO_4$ solution in ultrapure
deionized water to produce droplets that passed through a diffusion dryer and a neutralizer
before they entered the chamber. These seed particles served as a condensation sink for
condensable vapors in order to reduce vapor wall losses. To form OH radicals during
oxidation experiments we added nitrous acid (HONO) to the chamber by bubbling filtered
air through a HONO solution for 20 minutes.
We measured gas-phase organic species using both a proton-transfer-reaction mass
spectrometer (PTRMS, Ionicon Analytik) and a gas chromatograph/mass spectrometer
(GC/MS) (Agilent, 6890 GC/5975 MS) equipped with a thermal desorption and injection
system (TDGC/MS, Gerstel, MA) and a capillary column (Agilent HP-5MS, 30 m × 0.25
mm) (Zhao et al., 2014). We maintained the temperature of the PTRMS inlet line at 60 $^{\circ}$C
to minimize line losses. For the thermal desorption GC measurements, we collected
samples by drawing chamber air through Tenax® TA filled glass tubes (Gerstel 6mm OD,
4.5mm ID glass tube filled with ~290 mg of Tenax TA) at a flow rate of 0.5 L $min^{-1}$ for 2
minutes. We tracked the recovery of organics during analysis using C12, C16, C20, C24,
C30, C32, C36 deuterated n–alkanes as standards that we spiked into each Tenax tube prior
to the thermal desorption.



We measured particle number and volume concentrations inside the chamber using a
scanning mobility particle sizer (SMPS, TSI classifier model 3080, CPC model 3772 or
3010). We measured size-resolved and bulk particle composition and mass concentrations
with a high-resolution time-of-flight aerosol mass spectrometer (HR-ToF-AMS, Aerodyne
Research, Inc.). We operated the HR-ToF-AMS following the common protocol with the
vaporizer temperature at 600 ºC and electron ionization at 70 eV. We collected mass
spectra and particle time-of-flight (pToF) measurements in V-mode, which provides high
mass resolution (2000 m/$\Delta$m) and excellent transmission efficiency. We analyzed the AMS
data using the SQUIRREL V1.53G and PIKA 1.12G.
**3 Results and Discussion**
**3.1 Correction for the loss of the precursors, pinanediol, to the Teflon chamber walls.**
Because SVOCs should sorb to the Teflon walls, we expect a portion of PD to be lost after
PD was injected into our chamber. To constrain this, we injected equal quantities of six
compounds into our chamber simultaneously: PD, oxy pinocamphone, *n*-tridecane, 1-
tridecene, 2-nonanone, and 2-nonanol. The first two are an SVOC and an IVOC, while the
last four are VOCs that should have very limited wall partitioning at equilibrium. We then
measured the resulting gas-phase concentrations in the chamber using both TD-GC/MS
and PTRMS and compared the observed signals to those we expected based on the injected
amounts.
In Fig 1. we compare the TD-GC/MS measurements with the amounts of organics we
injected. The VOCs, *n*-tridecane, 1-tridecene, 2-nonanone and 2-nonanol, all fall along the
1:1 line, demonstrating that they have minimal wall losses and excellent recovery,



consistent with our expectations. However, PD and oxy pinocamphone show large
discrepancies between the measured and injected amounts. The recovered gas-phase values
show that 43% of the injected oxy pinocamphone and 86% of the PD were lost; only 14%
of the PD remained in the gas phase.
In Fig 2. we show the results of an experiment where we injected a succession of aliquots
of 1-tridecene, 2-nonanone, oxy pinocamphone and PD into the chamber, with expected
stepwise incremental increases of 11 ppbv each, and measured the gas-phase
concentrations with a PTRMS. We observed that the PTRMS signal stabilized after each
injection, and each injection with the same amount of organics resulted in a similar step-
wise vapor concentration increase. The two VOCs, 1-tridecene and 2-nonanone, both
showed concentration increases consistent with expectations. The PTRMS sensitivity to
nonanone is higher than its sensitivity to 1-tridecene, and so the signal to noise is
substantially higher. The 2-nonanone shows nearly square-wave response with a brief (~ 1
min) overshoot related to the chamber mixing timescale, and the 1-tridecene signal
displayed the same behavior. Oxy pinocamphone and PD show lower than expected
stepwise increases in concentration with a longer rise time. The step-wise increases for oxy
pinocamphone and PD are consistent with near constant wall-loss factors in the
concentration range in this study, but the signals are not consistent with instantaneous
evaporation and subsequent wall partitioning. If that was the case we would expect a large
initial spike similar and equal in magnitude to the spike in 2-nonanone (i.e. we would
expect the full 11 ppb to appear initially in the gas phase); we would then expect the SVOC
signal to drop to an equilibrium value on the equilibrium timescale for wall interactions –
10-15 minutes for our chamber (Ye et al., 2016a), as observed by Krechmer et al using a





core-flow inlet CIMS and nitrate chemical ionization (Krechmer et al., 2016). The slow
increase in signal we observe may be the convolution of two effects: less than instantaneous
evaporation from the flash vaporizer for the SVOCs and slow equilibration of the PTRMS
sampling line. Regardless, the signals in the PTRMS stabilize to values consistent with the
TD-GC/MS results; these experiments are both consistent with relatively rapid, reversible
equilibration of SVOCs (represented by the PD) and IVOCs (represented by the oxy
pinocamphone) between the gases and the Teflon chamber walls.
In order to calculate SOA mass yields, we must determine the amount of precursor oxidized
based on the change in precursor signals (e.g. the gas-phase PTRMS measurements). This
is straightforward for a VOC with minimal wall interactions, but for the SVOCs we must
account for their significant interaction with the Teflon walls. It is not sufficient to simply
measure the change in the gas-phase PD concentration, because of the apparently rapid
equilibration suggested by the theory put forward by Matsunaga and Ziemann and
supported by our wall-loss experiments. If PD were in equilibrium with the walls there
would be a substantial source of PD to the gas phase from the Teflon walls as PD was lost
from the gas phase due to oxidization or any other sink. Simply put, the results suggest
that, at equilibrium, for every 10 units of PD in the gas phase, roughly 100 units are sorbed
in or on the Teflon walls.  Therefore, removal of a small amount from the gas phase (say 1
unit) should result in replenishment of 90% by the walls to maintain the equilibrium.
Consequently, if we observe a decrease of 1 unit of PD vapor, that implies that 10 units are
actually lost from the gas phase since the evaporation of PD from the Teflon walls re-
establish the equilibrium. This, obviously, has large implications for the calculated SOA
mass yields above and beyond any possible wall losses for products of the PD oxidation.



We use two methods, heating and isothermal dilution, to test whether the Teflon chamber
walls in fact serve as an accessible reservoir of PD. Increasing the chamber temperature
raises the saturation concentration of PD and thus decreases the activity of PD vapors.
Heating by 30 $^{\circ}$C should raise the saturation concentration of PD by a factor of 10 to 30
and lower the gas-phase activity (the concentration divided by the saturation concentration)
by the same factor. Some PD sorbed to the Teflon should then evaporate to lower the
condensed-phase activity. To test this, we injected 866 µg m$^{-3}$ (118 ppbv) of PD vapor into
the chamber at 13 $^{\circ}$C and subsequently increased the chamber temperature to 44 $^{\circ}$C. As
shown in Fig. 3, the PD vapor concentration measured by the PTRMS increased rapidly
after heating and reached a steady value after the temperature stabilized at 44 $^{\circ}$C. The
concentration rose by a factor of 2.5-3. To be certain that desorption from the walls was
the only possible source, we also monitored the suspended aerosol mass using an HR-AMS.
The total organic mass in particles was around 5 µg m$^{-3}$, far less than the increase of the
PD vapor concentration. Particle evaporation thus contributed negligibly to the increase of
PD vapors; therefore, the PD adsorbed or absorbed by the Teflon chamber walls was the
only possible source of the increased gas-phase burden.
Increasing temperature by 30 $^{\circ}$C should increase the saturation concentration (C$^{*}$) of PD
by roughly a factor of 30 (May et al., 2012). All else being equal, this should cause a 30-
fold increase in the activity ratio of the sorbed PD to the gas-phase PD and thus drive a
large return flux to the gas phase, with the equilibrium vapor fraction increasing from 13%
to around 80%. This is consistent with our observations though we observe a factor of 2-3
less than this simple calculation would suggest. However, if PD is absorbed in the Teflon
walls, it is likely that the activity coefficient of the PD in Teflon walls would drop



substantially upon heating, so this would allow the activities to equilibrate with a smaller
net change in absolute concentration. Acknowledging these large uncertainties, the heating
experiment is broadly consistent with the postulated reversible equilibration of PD between
the gas-phase and the Teflon chamber walls.
Our SOA formation experiments are isothermal, but during the experiments the gas-phase
PD concentration (and thus activity) drops due to oxidation. To reproduce these conditions,
we used isothermal dilution to mimic the PD loss during SOA formation. We maintained
the chamber temperature at 22 $^{\circ}$C and injected PD along with acetonitrile into the chamber,
and then measured their concentration ratio using the PTRMS. We used acetonitrile as a
passive tracer because it is highly volatile, should not have wall losses, and it is readily
measured with the PTRMS. After injecting PD and acetonitrile into the chamber, we turned
on a slow flow of dilution air, initially at a rate of 100 Lpm (1% min$^{-1}$) and later at a rate
of 300 Lpm (3% min$^{-1}$). These rates roughly bracket the loss rate of PD via OH oxidation
in our SOA formation experiments. We tracked the ratio of PD to acetonitrile. If the PD
sorbed to the Teflon chamber wall were released continuously because it was in (a
necessarily reversible) equilibrium, the PD concentration should fall more slowly than
acetonitrile, and the ratio of PD to acetonitrile should rise steadily. We show a simulation
of the expected signals in Fig. S1. As we show in Fig. 4, the concentrations of both PD and
acetonitrile steadily decreased after we started to flush the chamber. However, we did not
observe any increase in the PD to acetonitrile ratio; instead, the ratio remained almost
constant, and even showed a slight decrease. This suggests that PD does not return to the
gas phase from the Teflon walls at 22 $^{\circ}$C, but instead still shows a modest loss to the



chamber walls. This indicates slow diffusion into the bulk Teflon, and is inconsistent with
the observations in Zhang et al. (Zhang et al., 2015).
During the dilution experiments, only after the PD concentration reached 2 µg m$^{-3}$ (2% of
the initial concentration), 5.5h after we started dilution, did the ratio of PD to acetonitrile
start to increase. This confirms that PD can return to the gas phase from the chamber walls
even during isothermal dilution (or any other isothermal loss from the gas phase), but only
after substantial depletion of gas-phase concentrations of PD. Thus, while reversible
partitioning to the walls is the most straightforward explanation for the losses of PD we
have presented, and even the results of chamber heating are broadly consistent with this
explanation, we see no sign of reversibility under the conditions of our SOA formation
experiments. This is a paradox, for which we have no explanation.
Therefore, based on the empirical evidence we conclude that the measured decrease in PD
from PTRMS during SOA formation experiments is equal to the amount of PD oxidation,
and that no further correction for wall equilibration is necessary. There is no reason for the
PD to "know" whether its gas-phase concentration is decreasing because of reaction or
isothermal dilution, and so we conclude that the dilution experiment accurately simulates
the PD response to reactive loss. However, as a precaution against return flux after
substantial PD depletion, we shall limit our analysis to the first 1.5 e-folding lifetimes in
PD oxidation (we only use the data where the PD concentration is above 22% of its initial
value).
**3.2 Correction for particle wall loss.**





We conducted experiments to measure the SOA production from oxidation of PD by OH
radicals generated via HONO photolysis at five different initial PD concentrations: 1, 2,
4, 5, and 6 ppbv. We used equation 1 to calculate SOA mass yields ($Y$).
$$Y = \frac{C_{SOA}}{\Delta C_{PD}} \tag{1}$$

where $C_{SOA}$ is the measured mass concentration of SOA, and $\Delta C_{PD}$ is the mass
concentration of the reacted PD. We measured the PD concentration using PTRMS with a
unique mass fragment, $m/z$=135, and then calculated the $\Delta C_{PD}$. As we have discussed, we
do not correct the measured concentration change in PD for any interaction with the
chamber walls. However, in order to calculate the $C_{SOA}$, we must also account for wall
losses of both particles and the condensable SOA products.
We employed three traditional methods to correct the particle wall loss, based on the
assumption that particles deposited to the chamber walls function as same as the suspended
particles for the SOA condensation. The corrected SOA production, $C_{SOA}$, is determined
by the ratio of suspended SOA ($C_{SOA}^{sus}$) to suspended ammonium sulfate seed ($C_{seed}^{sus}$) and
the initial concentration of ammonium-sulfate seed particles at time 0 h ($C_{seed}^{sus}(t = 0)$), as
shown in equation 2 (Hildebrandt et al., 2009).
$$C_{SOA}(t) = \frac{C_{SOA}^{sus}(t)}{C_{seed}^{sus}(t)} C_{seed}^{sus}(t = 0) \tag{2}$$

The essential term is the SOA to seed ratio, $\frac{C_{SOA}^{sus}(t)}{C_{seed}^{sus}(t)}$. We calculated this ratio directly from
the organic and seed (sulfate + ammonium) concentrations measured by the HR-AMS
(method 1). We also used the SMPS data. We determined the $C_{seed}^{sus}(t)$ by applying an





exponential function to fit the measured decay of the pure ammonium-sulfate seeds before
photo-oxidation and then extrapolating that decay for the duration of each experiment
(method 2). We also calculated $C_{seed}^{sus}(t)$ by scaling the total particle number concentration
(method 3). Because both coagulation and nucleation were minimal during the
experiments, **we can correct for particle wall losses based on either mass or number**
**loss**. $C_{seed}^{sus}(t)$ is proportional to the total suspended particle number concentration. We
demonstrate method 2 and 3 in Fig. S2. We calculated $C_{SOA}^{sus}(t)$ as the difference between
the total particle mass and the $C_{seed}^{sus}(t)$ after correcting with the SOA density, 1.4 g cm$^{-3}$,
which we calculated following the method of Nakao et al. (Nakao et al., 2013). As shown
in Fig. S3, the SOA to seed ratios from these three methods agree to within roughly 20%.
Consequently, we focused on the HR-AMS data (method 1) to perform the particle wall-
loss correction. We demonstrate one example of the temporal depletion of PD and SOA
formation in Fig. S4. Around 80% of PD reacted in the first hour. As mentioned previously,
we excluded all data where the PD concentration was less than 22% of its initial value from
the analysis; those data are plotted in gray.
**3.3 Correction for vapor wall loss.**
In addition to correcting for the loss of SOA as suspended particles, we also determine the
amount of condensable SOA vapors that condense directly to the Teflon chamber walls
after PD oxidation. This also reduces the observed SOA mass (Ye et al., 2016a; Krechmer
et al., 2016). If the condensing species are functionally non-volatile (their saturation ratios
are much larger than their particle-phase activity (Donahue et al., 2011b)), then
condensation to the suspended particles will be quasi-irreversible. Furthermore, for the
relatively low saturation concentration values required, there should be efficient wall losses





of the vapors. We thus assume that vapor wall losses are the same per unit condensation
sink as condensation to the suspended particles.
The condensation sink ($CS$) represents the loss frequency of vapors to the suspended
aerosol surface (Donahue et al., 2014); it can be thought of as the mean speed of the vapors
multiplied by the aerosol surface area, but modified for the gas-phase diffusion near the
particle surface and accounting for accommodation from the gas phase to the condensed
phase when that is rate limiting. We calculated the $CS^P$ using equation 3 (Trump et al.,

2014),

$$CS^P = \sum_k N_k \frac{v}{4} \pi d^2_{P,k} \beta_k \qquad (3)$$

where $k$ refers to a particle size bin, $N_k$ is the number concentration of particles in this

bin, $v$ is the mean thermal speed of the gas phase molecules, $d_{P,k}$ is the particle diameter,

and $\beta_k$ is the transition-regime correction factor (Seinfeld and Pandis, 2006), which is a

function of the mass accommodation coefficient (α) and the mean free path of the organic
vapor in air. We used two accommodation coefficient values, 0.1 and 1, as limiting cases

as the available evidence suggests that $0.1 < \alpha < 1$ (Saleh et al., 2013). When α = 1, the

condensation sink will be the same as the collision frequency between the gas molecules

and suspended particles.

Fig. 5 shows the suspended collision frequency versus time together with the number and
mass concentration of the suspended particles during an SOA formation experiment. The
collision frequency decreased initially due to particle wall losses. However, when the SOA
formation started, the SOA condensation increased the particle surface area and thus
increased the collision frequency. Later in the experiment, after the SOA formation was





almost complete, the particle wall loss again dominated and the collision frequency
decreased.
As shown in Scheme 1, the fraction of the oxidation products that initially condenses on
the suspended particles versus the chamber walls is determined by the ratio of the
suspended-particle condensation sink to the wall loss frequency (the wall condensation
sink). We previously measured a wall condensation sink for SVOCs in the CMU chamber
of 0.063 min$^{-1}$ (Ye et al., 2016a). In Fig. 6 we compare the suspended-particle condensation
sink to the wall condensation sink for the two limiting values of the mass accommodation
coefficient: 0.1 and 1. When $\alpha = 1$, the suspended-particle condensation sink is much larger
than the wall condensation sink. In this case, only a very small fraction of the condensable
vapors are lost to the walls, at least initially. When $\alpha = 0.1$, the condensation sink of the
suspended particles and the chamber wall are comparable, which makes vapor wall loss
significant.
The interactions of semi-volatile oxidation products with the two different sinks
(suspended particles and the walls) can be complex, but products that are effectively
nonvolatile (with very high steady-state saturation ratios while the PD is being oxidized
(Donahue et al., 2011b)) should simply condense in proportion to the two condensation
sinks. In this case the mass that condenses on the walls is given by the mass observed to
condense on the suspended particles multiplied by the ratio of the wall condensation sink
to the suspended condensation sink. In Fig. 7 we show the products lost to the chamber
walls together with the SOA mass on the suspended particles and the particles lost to the
chamber walls. The direct deposition of the product vapors to the chamber wall may have





been as much as 1/3 of the total SOA mass at the lower limit of $\alpha = 0.1$ or as little as a few
percent if $\alpha = 1$. This vapor wall loss correction is thus significant but not excessively large.

**3.4 Correction for Delayed Condensation.**

Some condensable products will be accumulated in the gas phase in a steady state between
production and loss even if they have a very low saturation concentration. This is especially
significant early in an experiment when the oxidation rate (and thus production rate of
condensable vapors) is high (Donahue et al., 2011b). We can estimate this simply by
assuming that the condensable vapors are produced with a constant mass yield during PD
oxidation (that the mechanism is invariant) and that their saturation concentrations are very
low. We then apply a constant mass fraction to the amount of oxidized PD to estimate the
total concentration of condensable products in any phase. In Fig. 8, we show an example
calculation for $\alpha = 0.1$ and a constant mass yield of 0.88 as a dashed black curve; except
for early in the reaction, this provides a good match to the total condensed organics, but for
times less than 2 condensation lifetimes (21 min, indicated with the vertical dashed red line)
the observed SOA concentration is substantially less than 0.88 times the oxidized PD
(shown with the gray fill). The SOA mass yields during the first 10-20 minutes thus may
be underestimated if delayed condensation is ignored (Donahue et al., 2011b). On the other
hand, lower mass yields at lower OA concentrations can be interpreted in terms of semi-
volatile partitioning (Odum et al., 1996b; Donahue et al., 2005).

**3.5 Overall SOA mass yields from PD oxidation by OH radicals.**

In Fig. 9 we show calculated SOA mass yields from the 6 ppb PD experiment for three
cases, first considering only particle wall loss, and then treating both particle and vapor



wall loss for α = 1 and for α =0.1. When α = 1, the difference with and without vapor wall
losses (i.e. the first two cases) is very small. However, the mass yield increases by 30%
after correcting for vapor wall loss with α = 0.1. We further estimate the delayed
condensation of ELVOC and LVOC products by finding the mass yield after two
condensation lifetimes, as illustrated in Fig. 8. The dashed horizontal lines indicate these
values. The true equilibrium SOA mass yields may be closer to the dashed lines than the
observed values due to delayed condensation.
In Fig. 10 we summarize data from five experiments with five different initial PD
concentrations: 1, 2, 4, 5, and 6 ppbv. The shaded area shows the range of SOA yields
when α values vary from 0.1 to 1. The instantaneous SOA mass yields are from 0.1 to 0.9
under the different SOA concentrations. As with the single case we present in Fig 9,
accounting for delayed condensation introduces a low-concentration asymptotic mass yield
between 0.4 and 0.8. The bottom line is that regardless of the mass accommodation
coefficient the SOA mass yields are high, with yields above 0.5 for $C_{OA} > 10$ µg m$^{-3}$. PD
oxidation by OH is thus a very efficient source of second-generation SOA.
The yields for α = 0.1 accounting for delayed condensation are implausibly high, implying
that essentially all of the oxidation products have extremely low volatility and thus the only
reason for the observed rising mass yields is the dynamical delay early in the experiment
(which lasts for a relatively long time, ~20 min, due to the low condensation sink associated
with the low mass accommodation coefficient). On the other hand, the yields for α = 1 are
plausible, implying that approximately half of the condensable oxidation products consist
of highly oxidized products formed via "auto oxidation" (Ehn et al., 2014) while the other
half are SVOCs that partition reversibly into the particles (Ye et al., 2016b; Ye et al., 2016c).



PD oxidation has much higher SOA mass yields than α-pinene oxidation. When $C_{OA}$ = 20
µg m⁻³, the SOA mass yields from α-pinene oxidation (by ozone or OH) are in the range
0.1–0.2 (Hallquist et al., 2009), whereas the SOA mass yields from PD oxidation by OH
are in the range 0.6–0.9, roughly five times larger. This finding holds regardless of wall
effects or other complications to quantitative interpretation of the product volatility
distribution, as those issues should be shared in common for each system. PD is a much
more effective source of SOA than α-pinene. This can be well explained by the structure
of PD. PD has two OH groups replacing the C=C double bond in α-pinene and yet it retains
the bicyclic backbone of that monoterpene. PD can be considered as a first-generation of
oxidation product of α-pinene; the likely atmospheric formation mechanism is hydrolysis
of a β-hydroxy nitrate formed after OH addition to the double bond in high-$NO_x$ conditions.
When PD is oxidized, C-C bond cleavage is unlikely because of the bicyclic backbone.
Therefore, most PD oxidation products will be less volatile than PD and so more
condensable compared to comparable products from α-pinene. One exception to this is that
a major oxidation product of PD is oxy-pinocamphone, which is formed when OH abstracts
a hydrogen atom from the hydroxymethylene moiety in PD and $O_2$ immediately abstracts
the second hydrogen from the OH group, analogous to acetone formation from 2-propanol.
All of the other oxidation products of PD are plausibly condensable. It is thus sensible that
the molar yields of condensable products from PD oxidation are in the range 0.5–0.8 and
that the corresponding mass yields are significantly higher due to the added oxygen.
**3.6 Elemental analysis of the SOA.**
In Fig. 11, we plot the observed average carbon oxidation state, $\overline{OS}_C = 2O:C - H:C$, of
the SOA formed from PD as a function of the SOA mass concentration. $\overline{OS}_C$ decreases as





the SOA mass increases, consistent with other studies of biogenic SOA (Donahue et al.,
2006; Shilling et al., 2009). The SOA that condenses very early in the experiment (at low
$C_{OA}$) is also highly oxidized. These promptly condensing organic products are ELVOCs or
LVOCs, with sufficiently low volatility to build up a high saturation ratio early in the
experiment. We also consistently observe a slight increase of $\overline{OS}_C$ at the end of each
experiment. This may be due to the further oxidation (aging) of the products. The SOA
formed from a lower initial PD concentration also shows a higher $\overline{OS}_C$ at the same SOA
concentration than the SOA formed from a higher initial PD charge. When the initial PD
concentration is low, the oxidation products may have more chance to react with OH
radicals and become more oxidized. However, it is also possible that the higher absolute
oxidation rate with higher PD concentrations drives up the gas-phase activity of SVOCs
with relatively lower $\overline{OS}_C$. Finally, it is possible that relatively more volatile (and less
oxidized) products are lost from SOA particles near the end of each experiment due to
sorption to the Teflon walls. As shown in Fig. S5, the ratio of organic to sulfate mass
decreased slightly after 2 hours, consistent with some SOA mass loss from the particles.
The composition findings are thus consistent with the mass-yield results for a relatively
high mass accommodation coefficient; there is a substantial mass yield of ELVOC and
LVOC products with very high $\overline{OS}_C$ but also a significant yield of SVOC products,
probably with $\overline{OS}_C \lesssim 1$, that dilute the (E)LVOC condensate once conditions favor their
condensation.
In Fig. 11 we also compare the $\overline{OS}_C$ of the SOA formed from PD in these experiments with
the $\overline{OS}_C$ of PD oxidation products observed to participate in nucleation in the CLOUD



experiment. We plot values for CLOUD for molecular clusters with a single $C_{10}$ molecule
and clusters with 4 $C_{10}$ molecules; these values are based on molecular formulas in
negatively-charged clusters measured with an atmospheric pressure interface time of flight
mass spectrometer (APITOF) where the negative charge resides on a bisulfate anion
clustering with the (presumably neutral) $C_{10}$ organic molecules formed from PD oxidation
(Schobesberger et al., 2013). The CLOUD values are thus based on a much different
technique than the highly fragmenting bulk particle electron ionization used in the AMS.
Despite these differences, the $\overline{OS}_C$ values we observe are similar to those seen in the
CLOUD experiments. The oxidized organics observed in the CLOUD experiments have
molecular compositions $C_{10}H_xO_y$, where x = 12, 14, 16 and y = 2–12 (Schobesberger et al.,
2013). They appear in four progressive bands from growing clusters, which contained 1-4
$C_{10}$ organic molecules, respectively. The $\overline{OS}_C$ in the first band is relatively high, -0.2, but
this decreases to -0.8 for the fourth band. The decrease of $\acute{OS}_C$ with increasing cluster size
is consistent with what we observed in this study. We observed the $\overline{OS}_C$ of the bulk SOA
at relatively high loading was around -0.7, which corresponds to the value measured in the
CLOUD experiments for larger clusters.
A self-consistent interpretation of these observations is that the least-volatile, early
condensing species forming SOA at low $C_{OA}$ in our experiments are ELVOCs that also
help form the smallest clusters in the CLOUD experiments, while the later condensing
species are LVOCs and SVOCs that also contribute to cluster growth in the CLOUD
experiment after initial nucleation.
**3.7 Representation of PD SOA in the two-dimensional volatility-oxidation space.**



Following the procedures in the literature (Presto and Donahue, 2006; Donahue et al.,
2011a), we mapped the distribution of volatility and $\overline{OS}_C$ in the two-dimensional volatility-
oxidation space (2D-VBS). The constraints are relatively crude – just the observed mass
concentrations and bulk composition, and so we present 2D-VBS yield distribution that is
consistent with those constraints but still coarse grained. Specifically, we assume a long
"tail" toward extremely low volatility with roughly constant mass yield, a cluster of
products with slightly lower volatility than PD, and a large yield of oxy pinocamphone,
while is more volatile than PD. We present the full yield distribution, which conserves
carbon, in the supplemental material.
In Fig. 12 we show the product distribution, classifying organics in the broad classes of
ELVOCs, LVOCs, SVOCs or IVOCs. The top panel is a 2D representation. We show PD
as a filled yellow circle. The blue contours show the oxidation products from PD, with
higher values indicating higher yields. The lower panel is a consolidation of the two-
dimensional product contours into a 1D-VBS, showing the total mass yields in each
decadally spaced volatility bin. A majority of the condensed products fall to the upper left
of PD, with a lower volatility and higher $\overline{OS}_C$ than PD. These compounds are produced
mostly by the addition of oxygen containing moieties to the PD backbone. However, some
products located on the right of PD show slightly higher $\overline{OS}_C$, but also higher volatility.
They may be formed by two possible reaction pathways. One is fragmentation, which
breaks the carbon backbone and produces smaller molecules with higher volatility than the
reactants. Another pathway is formation of oxy pinocamphone, as discussed above.
The products at the end of the low-volatility tail extending toward the upper left in the top
panel of Fig. 12 may contribute to the new-particle formation observed in the CLOUD

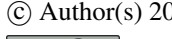



experiments. These ELVOCs, with log $C^o < -3.5$ are the most likely to form new particles
because with constant mass yields the saturation ratio in each progressively less volatile
bin will grow by an order of magnitude. The $\overline{OS}_C$ of these LVOC products ranges from 0
to 1, and they represent around 15 % of total SOA mass. This is consistent with CLOUD
observations showing that ~10% of the PD oxidation products could drive new-particle
formation (Schobesberger et al., 2013; Riccobono et al., 2014).
Employing the method of Chuang and Donahue (2016a), we conducted a dynamical
simulation of SOA production following oxidation of 6 ppb PD in the CMU chamber,
assuming a mass accommodation coefficient $\alpha = 1$. As shown in Fig. 13, the simulation
describes the formation of condensable vapors and subsequent production of SOA mass.
The suspended SOA mass in the simulation matches the smog-chamber data very well. The
particle mass and SOA vapors lost to the Telflon chamber wall are also comparable with
the calculated values from the experimental data. Especially during the first 15 minutes,
the simulation shows there is a large fraction of condensable SOA vapors in the gas phase.
This agrees with the observed condensation delay due to the condensation sink timescale.
**4 Conclusions**
Our studies show that oxidation of pinanediol, a semi-volatile surrogate for first-generation
oxidation products of monoterpenes, can produce SOA with very high mass yields. The
SOA is also highly oxidized. This is thus a model system to describe chemical aging of
first-generation SOA. Along with previously studied model systems for first-generation
products, this shows that aging of semi-volatile SOA is a significant source of additional
SOA mass, with higher mass yields typical of less volatile first-generation products. The
second-generation oxidation products with sufficiently low volatility represent 15% of the





total SOA mass in a 2D-VBS model that reproduces the chamber data; these may contribute
to new-particle formation. The oxidation state of the chamber SOA produced from
oxidation of PD is also consistent with the observations during new-particle formation
experiments at CERN. Thus, while first-generation oxidation is a substantial source of both
SOA mass and new-particle formation, ongoing oxidation of first-generation vapors, which
typically comprise the large majority of the first-generation oxidation products from
common precursors, should also be considered as a significant source of both particle
number and mass.
*Competing interests.* The authors declare that they have no conflict of interest.
*Acknowledgments.* This research was supported by grant AGS1136479 and AGS1447056,
from the National Science Foundation. The High-Resolution Aerosol Mass Spectrometer
was purchased with Major Research Instrumentation funds from NSF CBET0922643 and
the Wallace Research Foundation.





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

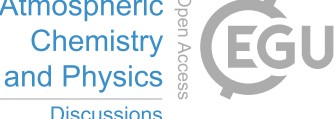



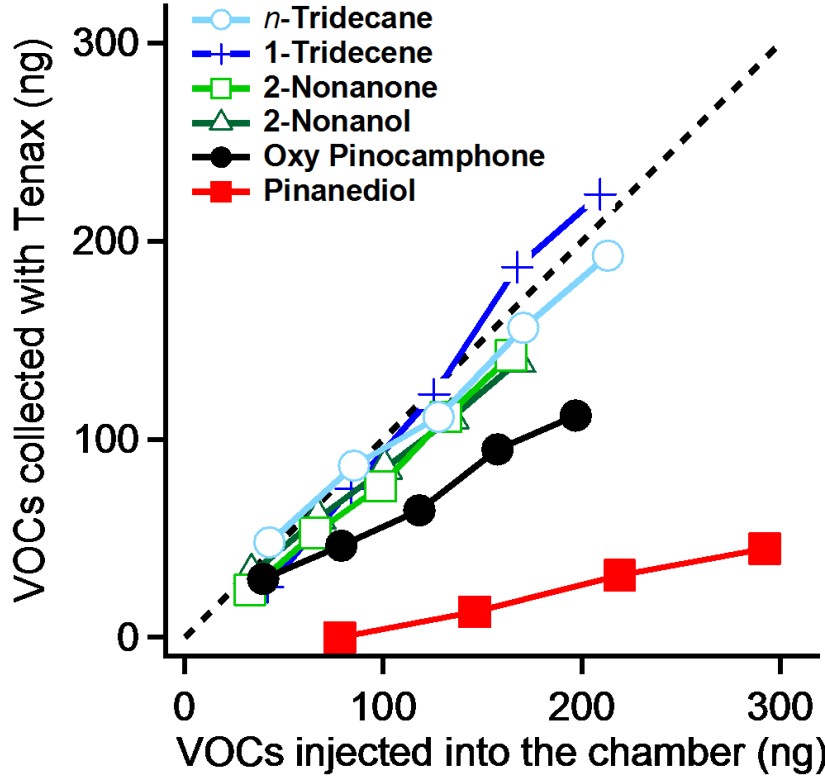


**Figure 1.** The gas phase concentrations of *n*-tridecane, 1-tridecene, 2-nonanone, 2-Nonanol, oxy pinocamphone and pinanediol in the chamber measured by TDGC/MS. Compared to the amount of organics injected into the chamber, *n*-tridecane, 1-tridecene, 2-nonanone and 2-nonanol show almost no vapor wall loss. Oxy pinocamphone and pinanediol show 43% and 86% loss, respectively.

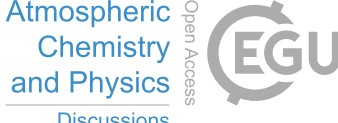

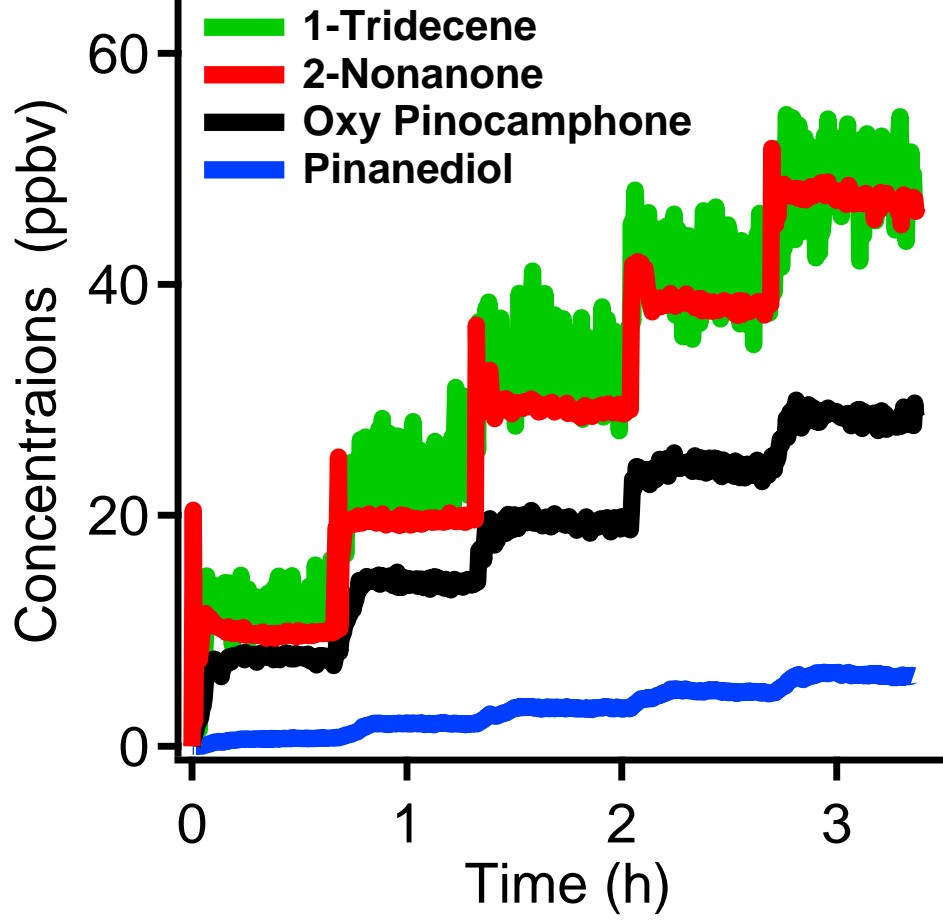

827

**Figure 2.** The temporal concentrations of the organics after we injected a series of aliquots of 1-tridencene,
2-nonanone, oxy pinocamphone and pinanediol into the chamber in increments of 11 ppbv (at 100% injection
efficiency). Each injection resulted in a similar increase of all organics. The similar increase indicates that
oxy pinocamphone and pinanediol may have constant wall loss factors in the concentration range studied in
this work.





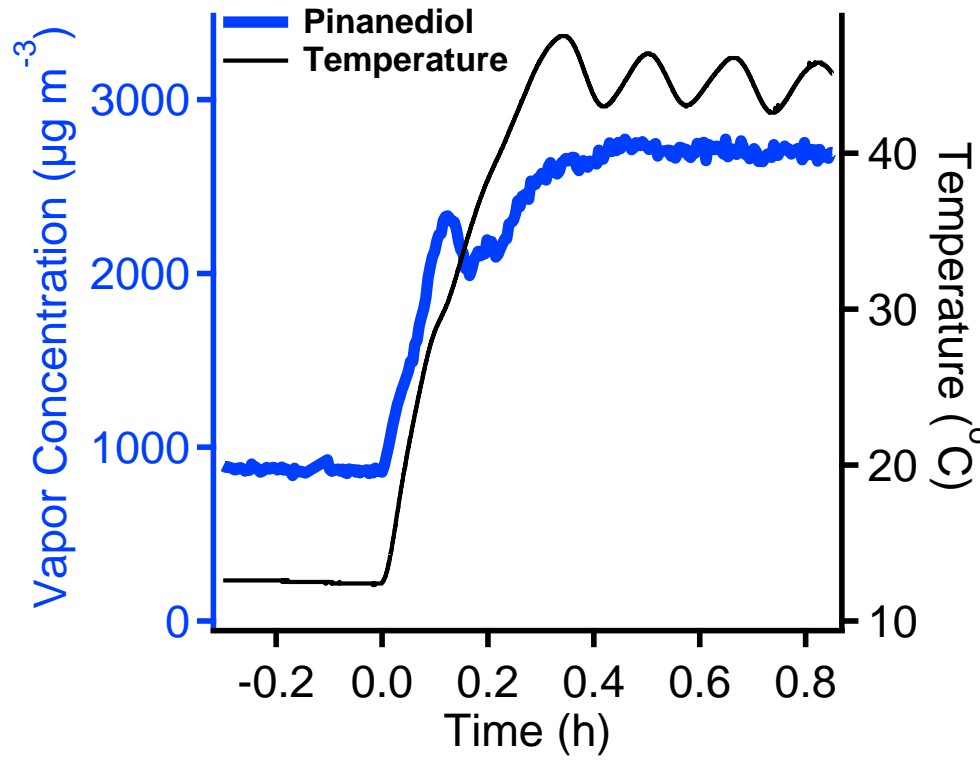

833

**Figure 3.** The increase of the pinanediol vapor concentration after increasing the chamber temperature from
13 ºC to 44 ºC. The concentration of PD increased 2.5-3 times and reached a constant value after temperature
stabilized at 44 ºC. The increase of the PD concentration shows that PD can come out from the chamber walls
at higher temperature.





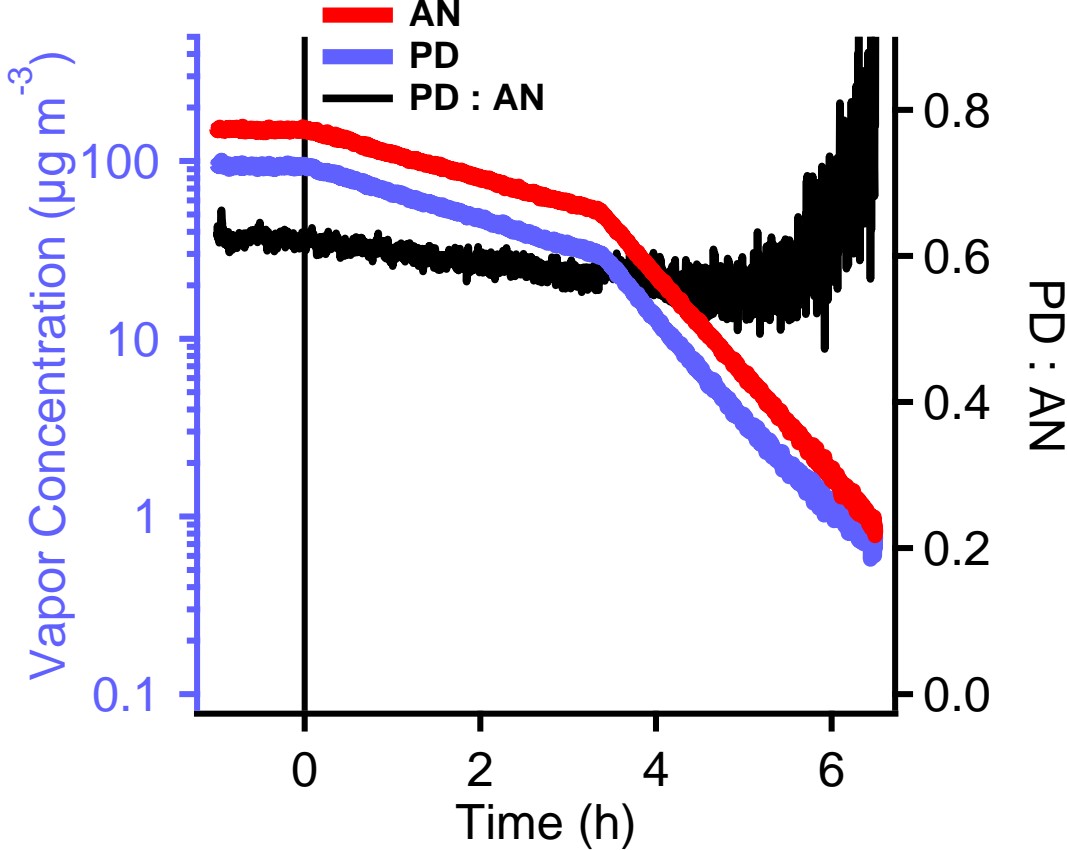

838

**Figure 4.** The change of PD and AN concentrations during isothermal dilution of the chamber with fresh air,
which mimics the depletion of PD during the SOA formation. The ratio of PD to AN shows very small change
until the PD concentration dropped below 2 µg m$^{-3}$. This indicates that PD does not return to the gas phase
from the Teflon at 22 °C, but instead still shows a modest loss to the chamber walls. So no further correction
for the release or loss of PD is necessary when studying the SOA formation.



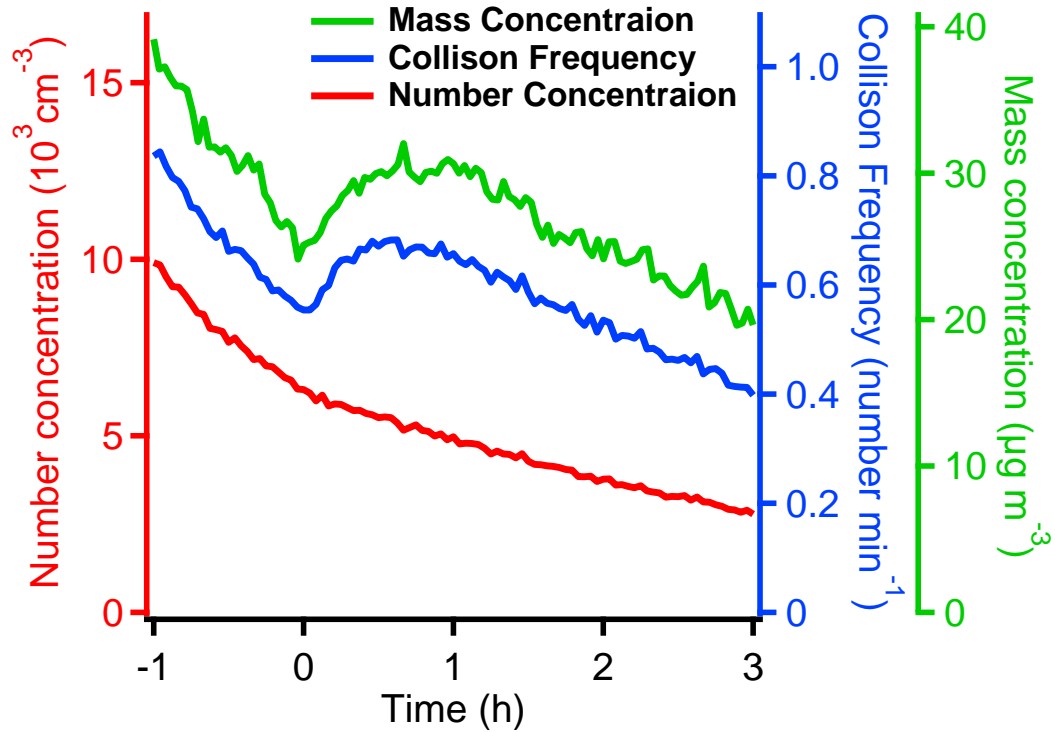

844

**Figure 5.** The change of collision frequency and number, mass concentration of the suspended particles
during the SOA formation. The collision frequency has the same value as condensation sink when α=1. After
the SOA formation started at 0h, the SOA mass condensed on the particles increased the particle surface
areas and increased the collision frequency. We also observed the increase of the total mass concentrations.
The particle number concentration always followed the exponential decay which indicated the nucleation
may be minimal.





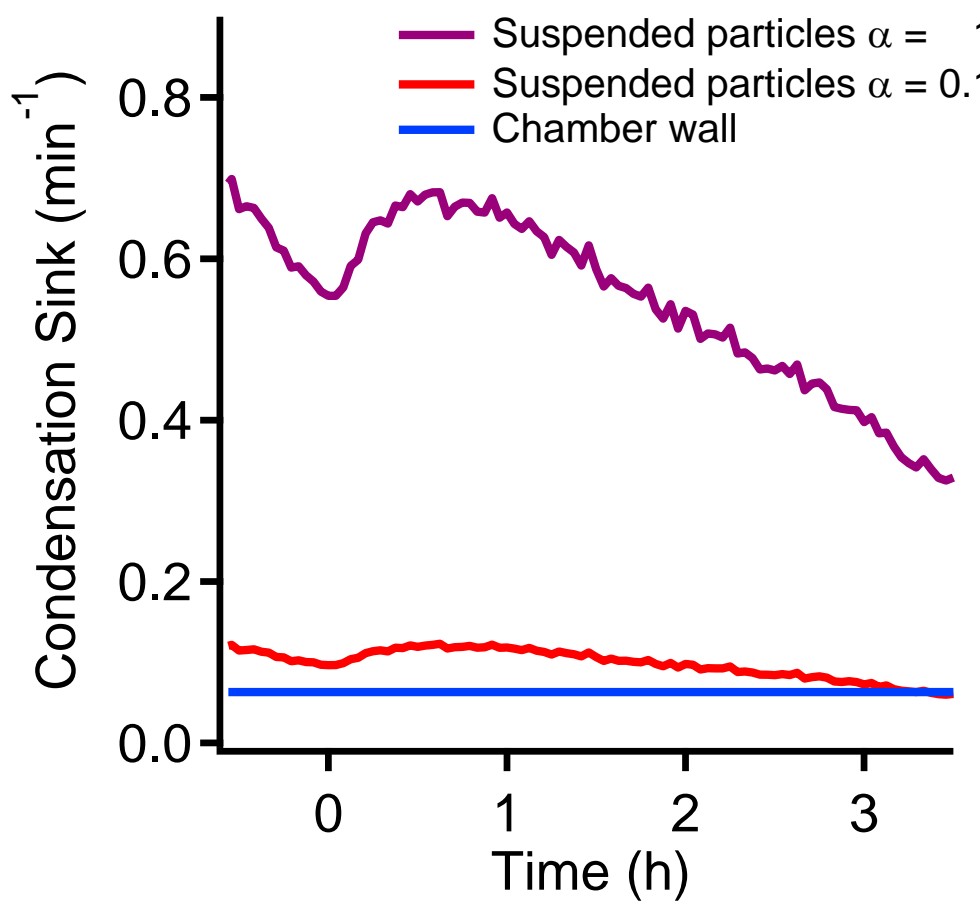

851

**Figure 6.** The difference of the condensation sink between the chamber wall with the suspended particles
when the mass accommodation coefficient is 0.1 or 1. When α = 1, the condensaiton sink of the suspended
particles is much larger than the wall condensation sink. When α = 0.1, the two values are on a similar level
which indicates that the vapor wall loss may be very significant.



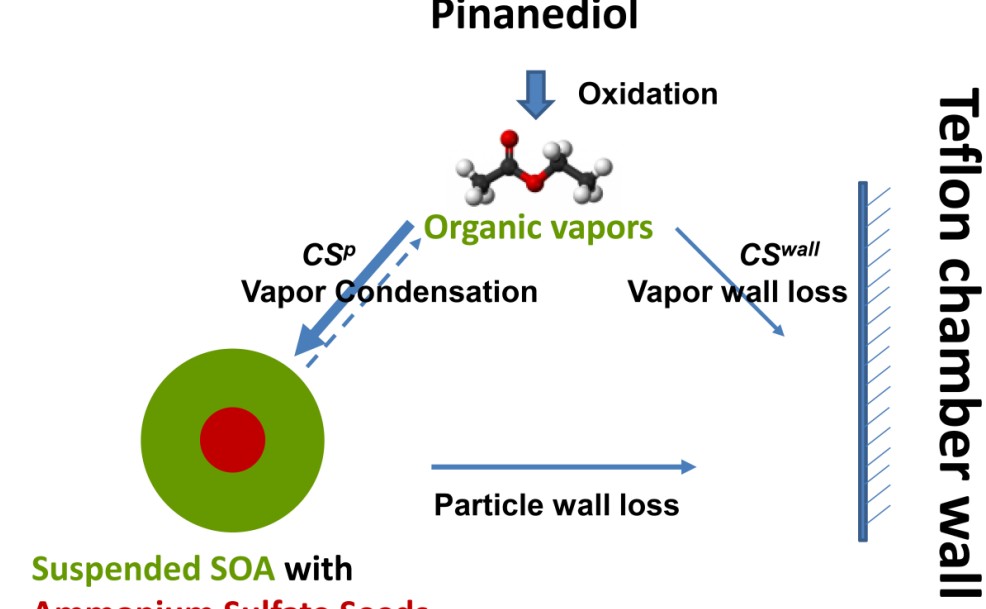


**Scheme 1.** The competition of vapor deposition on the suspended particles and the Teflon chamber walls.
The fraction of the oxidation products deposited on the suspended particles and the chamber wall are
determined by the condensation sink to the particles and the chamber walls





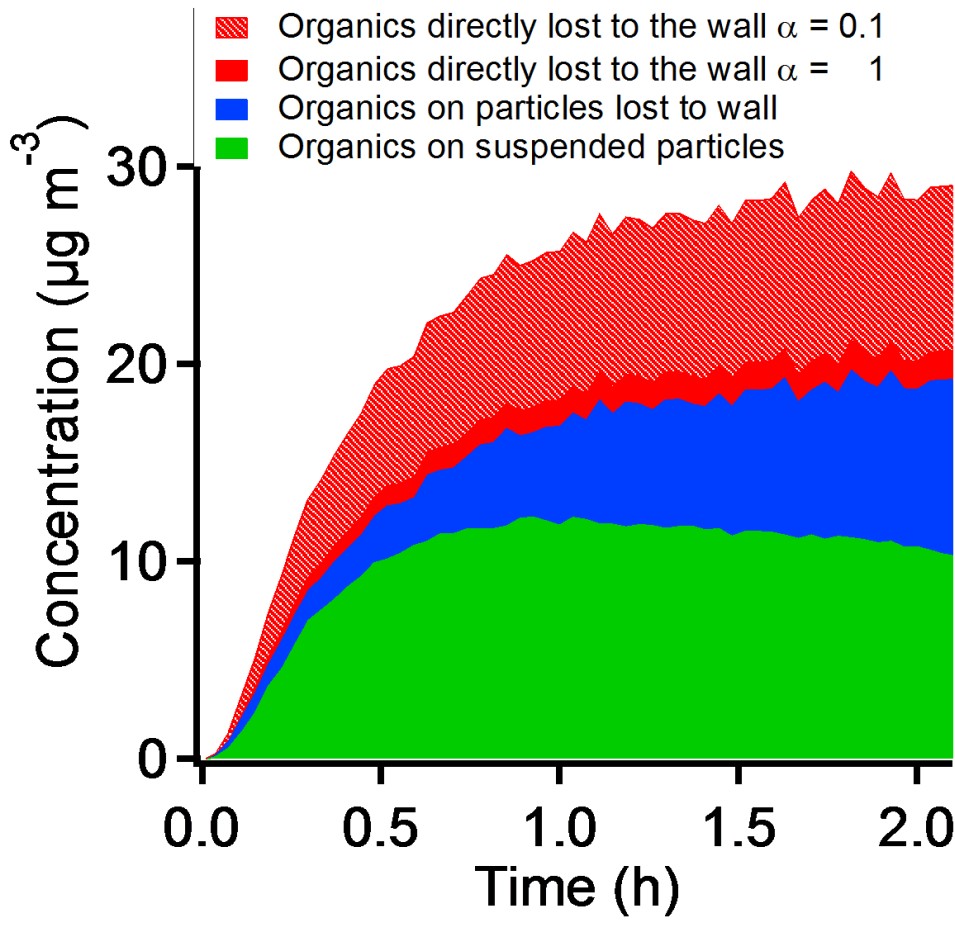


**Figure 7**. The SOA mass on the suspended particles, lost to chamber wall due to particle wall loss and direct vapor deposition on the chamber wall. When $\alpha = 0.1$, the SOA mass lost to the chamber wall through the direct vapor deposition may have one third of the total SOA mass. When $\alpha = 1$, the vapor wall loss may not be significant.





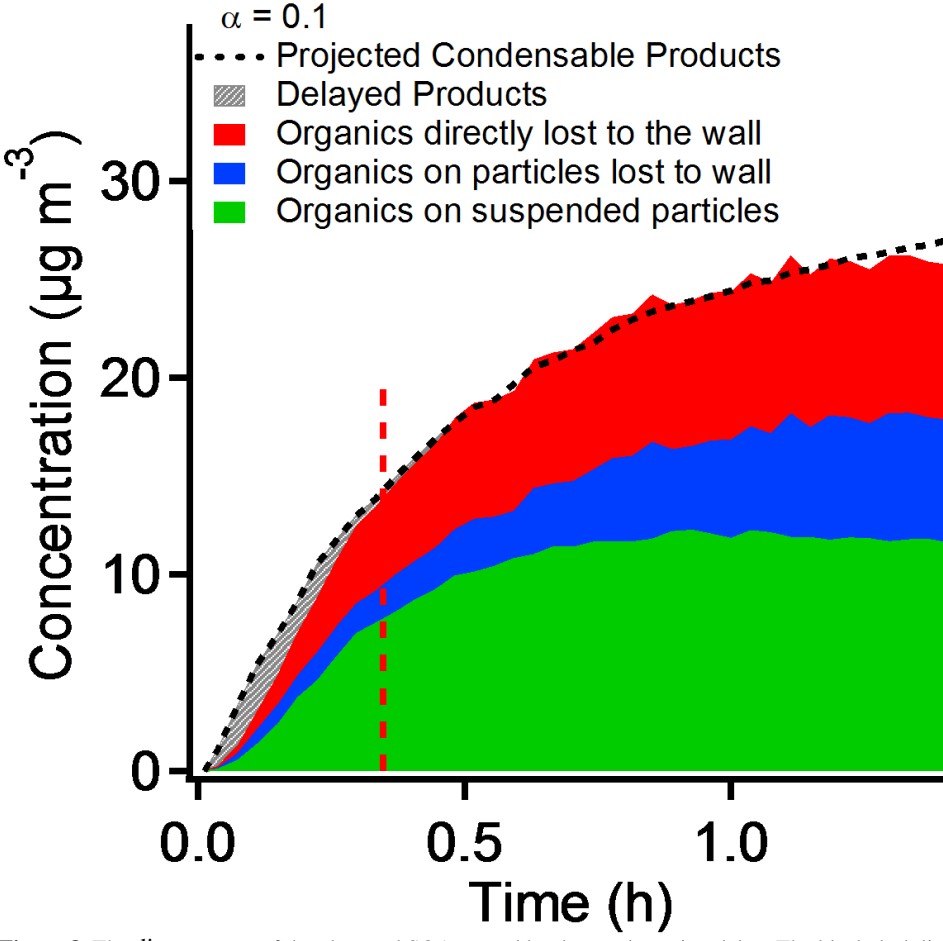

**Figure 8.** The discrepancy of the observed SOA caused by the condensation delay. The black dash line
shows the estimated concentration of condensable vapors from the reacted PD. The dashed area at 0-0.3
hours shows the difference between formed vapors and the observed SOA. This gap may be caused by the
diffusion time of vapor molecules to reach the surface of the particles or the chamber walls. This delay may
result in a lower measured SOA mass yield at the early stage of the experiment.





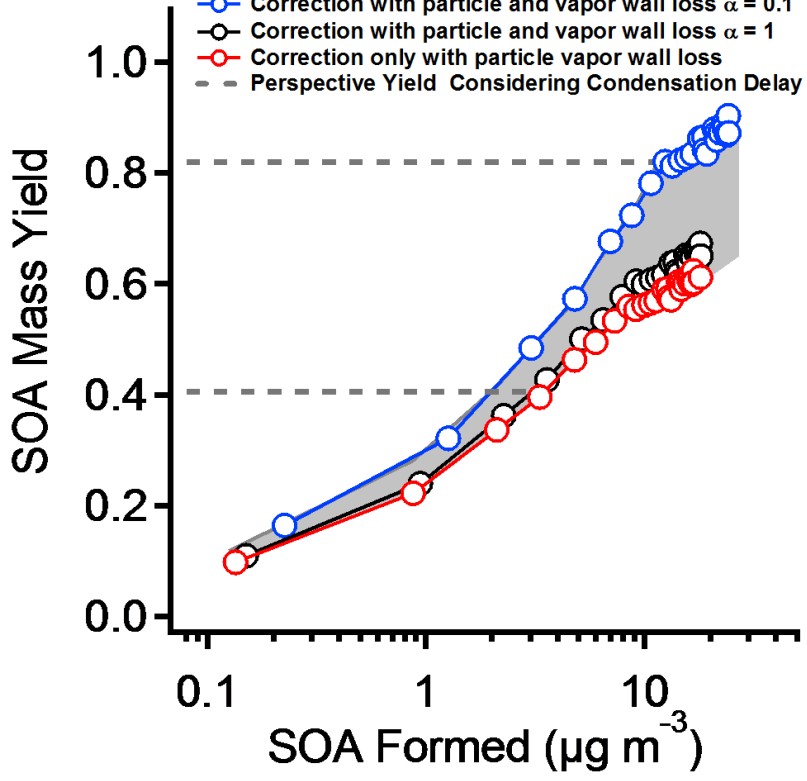


**Figure 9.** The SOA yield from pinanediol photo-oxidation after correction for particle wall loss and vapor wall loss using three different methods: correction for particle wall-loss only; correction for vapor wall loss with $\alpha = 1$; and correction for vapor wall loss with $\alpha = 0.1$. For the first two methods the mass yields are similar. For the third, when $\alpha = 0.1$, the mass yield is 30% higher than for the other two methods. The horizontal dashed lines indicate the mass yields at a time equal to twice the gas-phase lifetime of vapors due to condensation or wall loss. Before this time (below the lines) the measured SOA yields may be biased low due to the delay between production and condensation to the suspended particles.




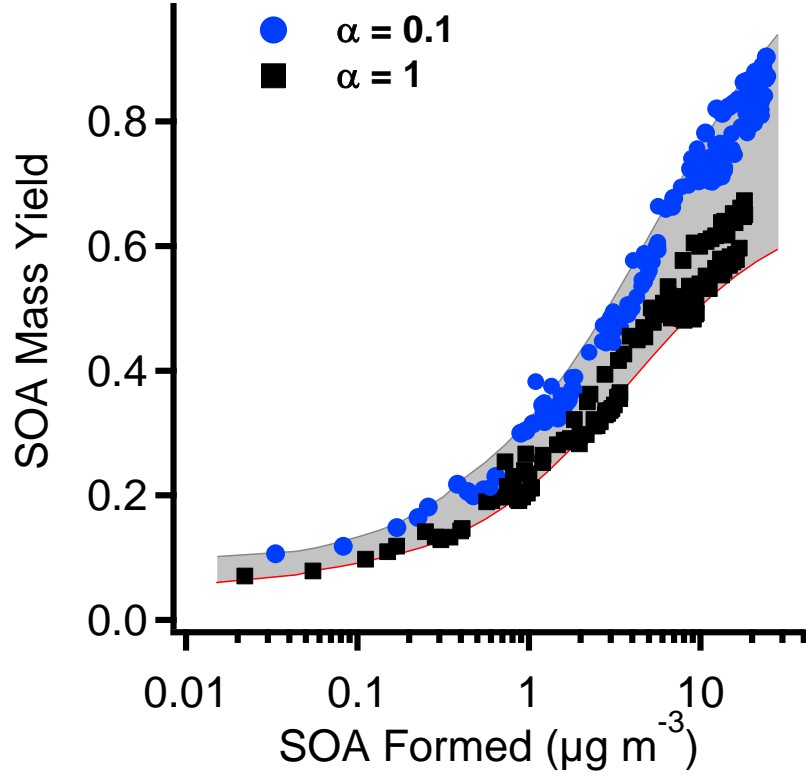


**Figure 10.** The summarry of all the SOA mass yield after correcting both particle and vapor wall loss. The
initial PD concentrations are 1,2,4,5, and 6 ppbv.The shade area shows the yield range when α varies from
0.1 to 1.




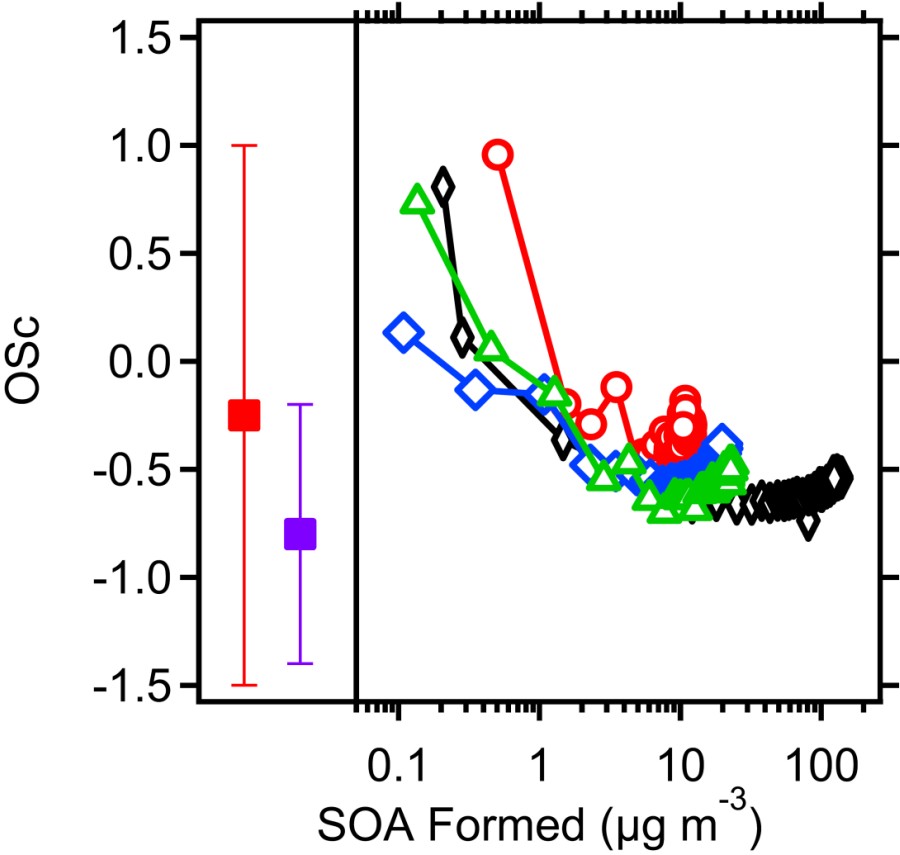


**Figure 11.** The $\overline{OS}_C$ of the SOA from PD with initial concentrations at 4, 5, 6 and 12 ppb on the right panel. The left panel shows the $\overline{OS}_C$ of the oxidation products from PD in the clusters observed in the CLOUD experiments, which contained 1 (red solid square) and 4 (blue solid square) $C_{10}$ organics. The SOA formed at the very early stage (low yields) shows highly oxidized. The $\overline{OS}_C$ in this study are comparable to the results from the CLOUD experiments.





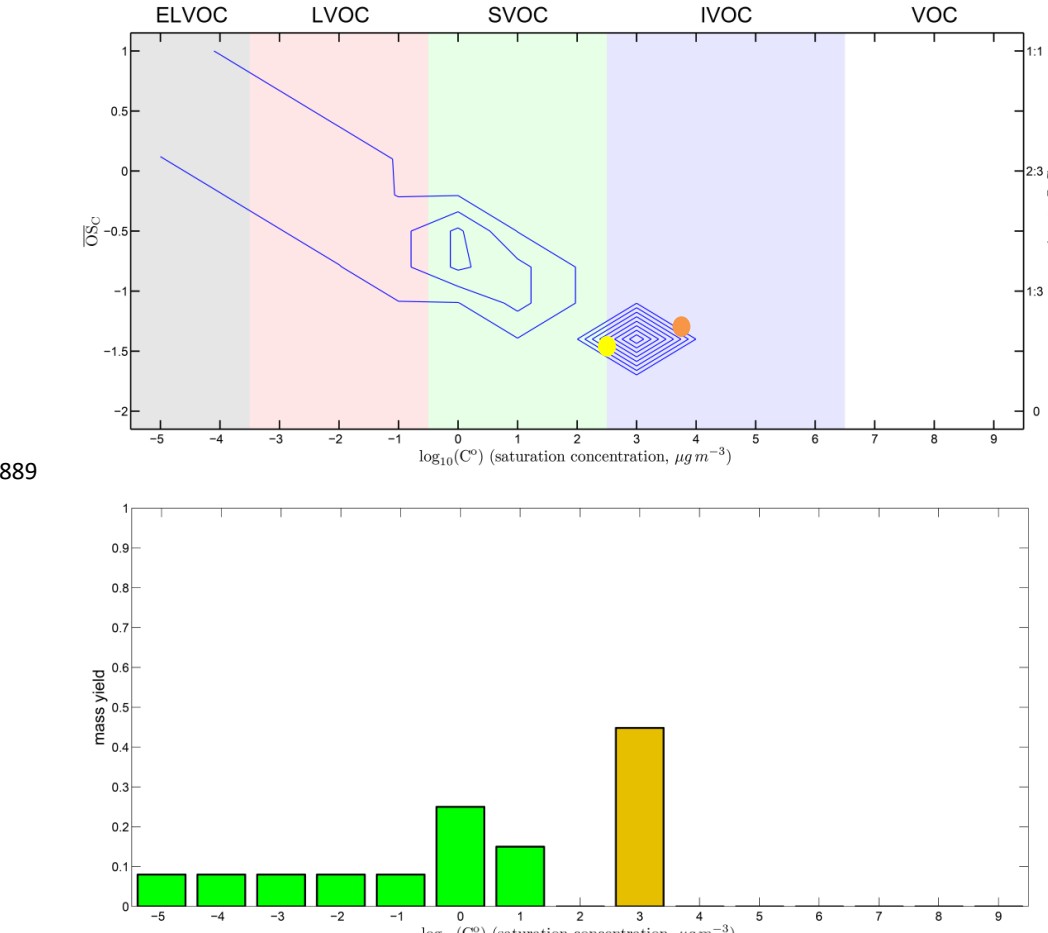


**Figure 12.** Representation of the oxidation products from PD in the two-dimensional volatility-oxidation
space for a mass accommodation coefficient $\alpha = 1$. We group organics in the broad classes of ELVOCs,
LVOCs, SVOCs or IVOCs. The top panel is a 2D representation. PD is shown as a yellow dot. The blue
contours show the oxidation products from PD, with higher values indicating higher yields. The lower panel
is a 1D consolidation of the 2D product contours, showing the total mass yields in each volatility bin. The
major products spread toward the upper left from PD, with increased oxidation state and decreased volatility.
The products near to the upper left corner, in the ELVOC region, may contribute to new-particle formation
observed in the CLOUD experiments. They constitute around 15% of the total SOA mass. Some products
may undergo fragmentation or functional group change, such as converting an alcohol group to a carbonyl
group, as with oxy pinocamphone, which is shown in orange.

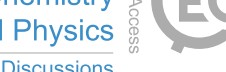

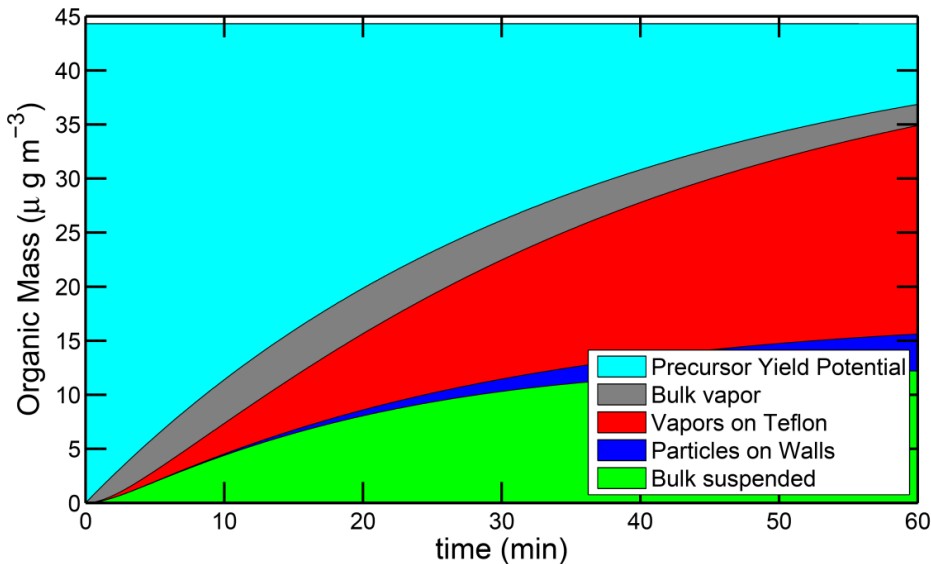

**Figure 13.** Dynamical simulation of the SOA production from 6 ppb of PD with a mass accommodation
coefficient α=1. The simulation treats five different reservoirs: unreacted precursor, vapors, suspended
particles, deposited particles, and sorption to teflon, as shown in the legend. The similation reproduces the
SOA observed on the suspended particles.