# Peer review of "Secondary organic aerosol production from pinanediol, a semi-volatile surrogate for first-generation oxidation products of monoterpenes"

_Atmospheric Chemistry and Physics, 2017_

## Referee Comment (RC1) · Anonymous Referee #1 · 25 Dec 2017

This study investigates the secondary organic aerosol (SOA) formation from photooxidation of pinanediol, chosen as a semi-volatile surrogate for first-generation oxidation products of monoterpenes. The authors found that the derived SOA mass yields, by accounting for vapor and particle wall losses, are 2-3 times larger than those from the oxidation of volatile monoterpene systems. By modeling the chamber data using a 2-D VBS set, the authors suggest that a significant fraction of pinanediol SOA comprises low volatility compounds, which is consistent with previous observations. Overall, the data analysis is thorough and the manuscript is clearly written and merits publication in ACP. Below are a few suggestions that the authors need to take into consideration for the production of a revised version of the manuscript.

General:

1. PTRMS calibration

The authors used PTRMS measurements to calculate the amount of pinanediol oxidized by OH radical in order to derive the mass yields of SOA produced. A recent study (Pagonis et al., AMT, 2017) has found that gas-wall partitioning of semi-volatile organics in Teflon tubing and inside the PTRMS could cause significant delays (up to two hours) in instrument response to step-function changes in the concentration of the semi-volatile compounds being measured.

As shown in Fig. 2, the authors in this study may have observed similar PTRMS response to the step-wise increases in the injected pinanediol in the chamber. This observation points to a very important factor that might lead to a large uncertainty in the calculated SOA yields, i.e., PTRMS calibration. The authors are suggested to describe in details how exactly PTRMS sensitivity to pinanediol was determined. If pinanediol standard was used, how was the vapor concentration calculated, and how was the vapor wall loss in the instrument accounted for?

2. Dilution experiments

Although the authors state that rapid gas-wall equilibrium partitioning of pinanediol (10-15 min?) was achieved in the chamber, no evidence could be found throughout of the manuscript. On the other hand, based on what is shown in Figure 2, it seems like there is a slowly decreasing trend in the measured concentration following each pinanediol injection. How did the authors define exactly the time it takes to reach gas-wall equilibrium partitioning?

The authors attribute the missing spike and the slow increase in the pinanediol signal upon a succession of standard injection to the slow equilibration of the PTRMS sampling line. This might also be the reason for the observed PD:AN ratio during the dilution experiment. How long does it take between the PD/AN injection and the onset of dilution? Is it possible that the PTRMS sampling line was far from equilibration with the pinanediol vapor in the sampling air during the entire dilution experiment (or at least the very first few hours)? If this is the case, then the sampling line could possibly act as a constant sink of the pinanediol vapor and the amount evaporated from the wall upon dilution of the chamber might be compensated by that deposited onto the sampling tubing. Have the authors thought about why the PD/AN ratio only started to increase after 5 hours

of dilution (or the PD concentration dropped below 2% of the initial concentration?) This gas-wall partitioning behavior seems very inconsistent with the observation from the heating experiment.

3. Vapor wall loss correction

The authors used a single wall condensation sink (0.063 min$^{-1}$) measured for SVOCs in the CMU chamber to account for wall losses of vapors across all the volatility range, including LVOCs. While the time for establishing gas-wall equilibrium might be similar (say 10-15 min) for different organic vapors, it has been shown, by many studies, that the amount of organic vapors that reside in the chamber wall phase upon equilibrium depends on the vapor pressure (e.g., Matsunaga and Ziemann, 2010; Zhang et al., 2014, Krechmer et al., 2016). Here by comparing the vapor condensation rate to the wall vs. particles to evaluate the underestimation of SOA yields due to vapor wall loss may bare large uncertainties, as the amount of organic vapors in the wall upon equilibrium partitioning as a dependence of vapor pressure is not accounted for.

4. Accommodation coefficient

The accommodation coefficient is widely used to represent the probability of a vapor molecule sticking onto an organic particle surface. However, the accommodation coefficient used in Equation 3 in this study is essentially an effective accommodation coefficient, as the particle-phase diffusion process needs to be accounted for. Many studies have found that under dry conditions, the phase state of a-pinene SOA is more like semi-solid, implying that the particle-phase diffusion might be the rate limiting step in the overall gas-particle partitioning process. Please comment on the range of accommodation coefficient (0.1-1) chosen here.

Minor:

1. Line 211: Specify how long it takes between the chemical injection into the chamber and the measurement of their concentrations by PTRMS/GCMS. What is the chamber mixing timescale?

2. Line 252: Please show evidence for the 'rapid vapor-wall equilibrium' observed in the experiments.

3. Line 295: Again, specify the time duration between chemical injection and the onset of chamber dilution.

---

## Referee Comment (RC2) · Anonymous Referee #2 · 4 Jan 2018

Ye et al. present a chamber study of secondary organic aerosol (SOA) formation from pinanediol (PD), a proxy of a first-generation product of monoterpenes. The authors give careful attention to the wall loss of both particles and vapors, and, with this accounting, the SOA yields are found to be 2-3 times larger than those of typical monoterpenes. Analysis of dynamical 2D-VBS modeling suggests the formation of a significant fraction of ELVOC from PD oxidation. The paper requires a number of clarifications before it can be published.

**General comments:**

1. What 8 $m^3$ chamber has a surface area of 12 $m^2$ (line 105)? This is off by a factor of probably about 2. The smallest surface area to volume ratio is that of a sphere, and a sphere with a volume of 8 $m^3$ would have a surface area of 19.3 $m^2$. Likely, any chamber with this volume would have an even larger surface area (and certainly much larger than 12 $m^2$). Related to this, what is the source of the estimate of 10 g in line 107?

2. How are the data points in Fig. 1 obtained, since in Fig. 2 there is a slightly decreasing trend when the concentrations reach "quasi-steady-state"?

3. How do you perform the stepwise injection of the compounds in Fig. 2, i.e. at each injection step does the volume of the chamber change because of constant sampling? Also, you mention the longer evaporation time of the less volatile compound: can you give an estimated timescale?

4. In the heating experiment (Fig. 3), how much PD do you inject into the chamber at 13$^o$C in order to get 866 $\mu$g $m^{-3}$? Have you tried to increase temperature to just 22$^o$C to see if you can get a similar portion of bulk concentration of PD with the ones in Fig. 2? In other words, how can you verify the possibility of pure condensation of PD on the wall or other lines at such a lower temperature? Otherwise, one would think the vapor-wall interaction mechanism is different in heating and dilution experiments.

5. In the dilution experiment, you show that PD-wall partition is irreversible above 22% of the initial value, which may be true if the oxidation rate of PD is similar to the dilution rate. So how do you simulate the photo-oxidation of PD? What are the actual values of $j_{HONO}$ and OH level in the chamber? What is the oxidation mechanism used in the simulation: parallel or in series?

6. If the conclusion in lines 318-320 is correct, why does Fig. 1 not have a y-intercept of 0? Also, how are you accounting for the additional loss you saw in the experiment for Fig. 4?

7. Around line 372, you are assuming that the condensation sink does not change as more vapor deposits throughout the experiment. How do you justify this assumption, particularly for the boundary layer? The mass transport through the boundary later is changing throughout the experiment, so the condensation sink of deposited particles also changes.

8. Can you clarify the necessity of the correction for delayed condensation? In the caption of Fig. 8, you attribute the delayed condensation to the diffusion time of vapor molecules to the surface of the particles or the wall. Do you mean the gas-phase production rate is too fast compared with the timescale to reach gas-particle-wall equilibrium, so that the instantaneous equilibrium assumption cannot be used at the initial stage?

9. Since you are comparing your experiment to a nucleation experiment in CLOUD (lines 513-520), you should justify your assumption that you used enough seed to suppress nucleation when discussing particle number concentration (line 351).

10. How do you distinguish "overall SOA yield" and "instantaneous SOA yield"? It looks like Fig. 9 and Fig. 10 are plots of temporal profile of overall SOA yield.

**Specific comments:**

Line 93: Remove the symbol "‡" in the citation.

Lines 91-94 repeat what is more succinctly said in line 89.

Line 163: I believe the unit is $m^3$ not $m^{-3}$.

Line 178: What type of neutralizer did you use?

Lines 199, 265, 269, 271, 306: There should not be a space before °C.

Lines 213 and 220: The period should go after "Fig" not after the number, as is done in the rest of the paper.

Line 218: Why does it look like the y-intercepts for oxy pinocamphone and PD are not

0?

Line 228/Fig. 2: The overshoot time for 2-Nonanone appears to be a lot closer to 10 minutes than to 1 min, especially for the data a little after 2 hours.

Lines 265 and 278-280: These sentences repeat each other but, in line 265, you say "factor of 10 to 30" and in lines 279-280 you just say "30-fold increase." What happened to the range in the second sentence?

Line 283: The PD should be "absorbed into" or "absorbed by" the Teflon walls, not "absorbed in" them.

Line 307: Does the ratio decrease before dilution when the concentration is held constant? Otherwise, diffusion into the bulk Teflon does not make sense.

Line 308: This is the wrong Zhang 2015 reference.

Line 310: There should be a space after "5.5" before "h," as is done in the rest of the paper.

Line 317: Did you try slowing the rate of dilution even more to see if there was an effect?

Line 339: "as same as" should be "the same as" or something of that sort.

Line 352: The font is bold.

Line 352: How do you verify ignoring other dependencies? E.g. the dependence of the wall loss rate on the diameter of the particle.

Lines 384 and 386: These lines have odd spaces/indentations.

Line 437: Inconsistent spacing after the equals sign.

Line 474: Be consistent between "oxy-pinocamphone" and "oxy pinocamphone."

Line 466-476: It is better to represent the chemical mechanism in a scheme.

Line 517: $OS_C$ needs a line above it instead of an accent mark.

Line 535: Where in the supplemental material is this provided?

Line 536: You should probably mention this is for α=1 and give the justification for choosing this value of α that you give in the figure captions.

Lines 561, 880, and 904: "Teflon," "summary," and "simulation" are misspelled.

Lines S26-S28: It is unclear when you switch to an explanation of method 3.

Figure 3: Why is there a bump/overshoot in the Pinanediol concentration around 0.1 hours?

Figures 4 and S1: Why not make these A and B parts of a figure, so that they can be more directly compared?

Figure 5: The SMPS used in this experiment cannot detect nano-particles, so the last sentence about nucleation may not stand.

Figure 7: Use another color or background for the case α = 0.1.

Figure 7: The solid red versus thickly shaded red are very difficult to distinguish, even when viewed in color.

Figure 8: Since you already use red in the figure, it may make more sense to replace the red dashed line with another color.

Figure 11: This figure is missing a legend.

Figure 12: Missing colorbar for contour lines.

Figure 13: I suggest you change "Bulk suspended" to "Particle suspended" in the legend.

Figures S2 and S3: Cn is never defined. Also, in S2, the labels on the blue arrows are sufficiently far away from these arrows to be somewhat confusing.

---

## Author Comment (AC1) · 14 Mar 2018

Reviewer 1:

General:

1. PTRMS calibration The authors used PTRMS measurements to calculate the amount of pinanediol oxidized by OH radical in order to derive the mass yields of SOA produced. A recent study (Pagonis et al., AMT, 2017) has found that gas-wall partitioning of semi-volatile organics in Teflon tubing and inside the PTRMS could cause

significant delays (up to two hours) in instrument response to step-function changes in the concentration of the semi-volatile compounds being measured. As shown in Fig. 2, the authors in this study may have observed similar PTRMS response to the step-wise increases in the injected pinanediol in the chamber. This observation points to a very important factor that might lead to a large uncertainty in the calculated SOA yields, i.e., PTRMS calibration. The authors are suggested to describe in details how exactly PTRMS sensitivity to pinanediol was determined. If pinanediol standard was used, how was the vapor concentration calculated, and how was the vapor wall loss in the instrument accounted for?

ANSWERS: We determined the PTRMS sensitivity to pinanediol by comparing the PTRMS signals with the pinanediol concentrations inside the chamber. We measured the pinanediol concentration using TD-GCMS. We collected samples by drawing chamber air through Tenax® TA filled glass tubes. We used pinanediol in methylene chloride solution with different pinanediol concentrations as the GCMS calibration standard.

Our sampling setup is different from the study Pagonis et al., AMT, 2017. We used a steel sampling tube and heated the line to 60oC. We wanted to minimize the loss of pinanediol to the sampling tube wall or inside the instrument. We found the PD signals dropped to near to zero immediately after we disconnected the sampling tube from the chamber.

2. Dilution experiments Although the authors state that rapid gas-wall equilibrium partitioning of pinanediol (10-15 min?) was achieved in the chamber, no evidence could be found throughout of the manuscript. On the other hand, based on what is shown in Figure 2, it seems like there is a slowly decreasing trend in the measured concentration following each pinanediol injection. How did the authors define exactly the time it takes to reach gas-wall equilibrium partitioning? The authors attribute the missing spike and the slow increase in the pinanediol signal upon a succession of standard injection to the slow equilibration of the PTRMS sampling line. This might also be the reason for the observed PD:AN ratio during the dilution experiment. How long does it take be-

[Figure]

Interactive
comment

tween the PD/AN injection and the onset of dilution? Is it possible that the PTRMS sampling line was far from equilibrium with the pinanediol vapor in the sampling air during the entire dilution experiment (or at least the very first few hours)? If this is the case, then the sampling line could possibly act as a constant sink of the pinanediol vapor and the amount evaporated from the wall upon dilution of the chamber might be compensated by that deposited onto the sampling tubing. Have the authors thought about why the PD/AN ratio only started to increase after 5 hours of dilution (or the PD concentration dropped below 2% of the initial concentration?) This gas-wall partitioning behavior seems very inconsistent with the observation from the heating experiment.

ANSWERS: The 10-15 mins timescale was calculated for SVOCs in the chamber in our previous paper (Ye et al., 2016a), and also observed by Krechmer et al (Krechmer et al., 2016). In Fig. 2, the slow increase was caused by three factors, the injection time (15mins), the chamber mixing time (5-10mins), and the gas wall partitioning equilibration time (10-15mins). These three factors overlapped each other and could not be determined individually. However, we have very strong evidence from both direct observations of $H_2SO_4$ vapor loss as well as SVOC loss from coated particles, as reported in Ye et al., 2016a, that the intrinsic chamber-wall collisional timescale is 10-15 minutes for compounds with the molecular weights of interest here, including analogues to PD such as oleic acid. Even the differences in timescales (10 min for $H_2SO_4$, 15 min for heavier organics) are consistent with theoretical expectations. PD has a higher vapor pressure than most of the SVOCs employed in Ye et al., 2016a, though it is near the high end of the range employed there. It would be very surprising if the PD equilibration timescale were significantly longer and impossible for it to be shorter (the vapor-wall collisional timescale is a lower limit).

We waited around one hour between the PD and acetonitrile (AN) injection and the onset of dilution. If PTRMS sampling line was far from equilibration with the pinanediol vapor in the sampling air, we should observe a very low signal during the injection followed by a steady increase for the hour before we started the dilution. We only observed the continuous decrease after dilution started. Given that both the PD and AN signals both dropped significantly during the dilution experiment (that was the point) and that we are very confident that the AN is a truly passive tracer in both the chamber and the PTRMS and its sampling line, it would take an extraordinary confluence of events for the ratio of the two signals to remain almost perfectly constant without that reflecting a true passive dilution in the chamber itself. It seems not likely that the PTRMS sampling line is far from the equilibration, and thus our conclusion is that the actual gas-phase concentration of PD in the chamber declined during dilution consistent with passive dilution and thus no return flux from PD absorbed or adsorbed to the chamber walls.

That being said, there is an obvious inconsistency in the complete set of observations; nothing can equilibrate without a balance of forward and reverse fluxes, and we have ample evidence of significant PD loss to the chamber walls that non-the-less resulted in a constant PD gas-phase signal proportional to the amount of injected PD. Those are the hallmark signatures of equilibration, as pointed out by Matsunaga and Ziemann. The heating experiments confirm that a large fraction of the PD did indeed partition to the walls. We are fully aware of the inconsistency here, and yet the scientific question of SOA formation from SVOCs in general and PD in specific is pressing. We are still trying to get a good explanation of the different gas-wall partitioning behavior between the dilution and heating experiments. One possible reason may be the evaporation energy of the pinanediol on the chamber wall. The evaporation rate became much higher after heating up the chamber. Then we observed the increase of the pinanediol concentration in the gas phase; however, this does not solve the evident inconsistency at room temperature. Consequently, we adopted the practical and empirical approach of using the dilution experiments as a controlled test to mimic PD loss via chemical reaction. In this way we are comfortable that we can constrain the total amount of PD oxidized during the experiment, which is absolutely essential for a mass yield determination, but in an abundance of caution we restricted our analysis to the period when at least 20% of the PD remained in the chamber (a factor of 10 more than the point where

the dilution experiments showed signs of disequilibrium).

3. Vapor wall loss correction The authors used a single wall condensation sink (0.063 min-1) measured for SVOCs in the CMU chamber to account for wall losses of vapors across all the volatility range, including LVOCs. While the time for establishing gas-wall equilibrium might be similar (say 10-15 min) for different organic vapors, it has been shown, by many studies, that the amount of organic vapors that reside in the chamber wall phase upon equilibrium depends on the vapor pressure (e.g., Matsunaga and Ziemann, 2010; Zhang et al., 2014, Krechmer et al., 2016). Here by comparing the vapor condensation rate to the wall vs. particles to evaluate the underestimation of SOA yields due to vapor wall loss may bare large uncertainties, as the amount of organic vapors in the wall upon equilibrium partitioning as a dependence of vapor pressure is not accounted for.

ANSWERS: The organics in the SOA are mostly SVOCs, LVOCs and ELVOCs. These organics equivalent saturation concentration in the wall upon equilibrium are more than milligrams/m3, which is far higher away for the concentration we used in this study. We also used seed concentrations high enough so that the collision timescale to the suspended seeds was more than an order of magnitude higher than the collision timescale with the walls, as discussed in the paper. These two things combined mean that the very large majority of condensable vapors (LVOCs and SVOCs) that encountered the walls should remain there (the equilibrium fraction was < 0.001) but also that most of the SVOCs and all of the LVOCs should have remained on suspended particles for at least a significant portion of the experiment (the other way to think of this is that the steady-state excess saturation between the gas phase and the particles was relatively high during PD oxidation, so the net flux to the suspended particles was close to that of a truly non-volatile constituent. For these reasons we modeled the loss of the SOA vapors to both the chamber walls and the suspended particles as quasi-irreversible. This is definitely an approximation, but our objective is to set up experimental conditions where we are not reliant on uncertain model parameters (i.e. the exact volatility

and wall partitioning constants) of condensable species.

4. Accommodation coefficient The accommodation coefficient is widely used to represent the probability of a vapor molecule sticking onto an organic particle surface. However, the accommodation coefficient used in Equation 3 in this study is essentially an effective accommodation coefficient, as the particle-phase diffusion process needs to be accounted for. Many studies have found that under dry conditions, the phase state of a-pinene SOA is more like semi-solid, implying that the particle-phase diffusion might be the rate limiting step in the overall gas-particle partitioning process. Please comment on the range of accommodation coefficient (0.1-1) chosen here. ANSWERS: The ELVOCs are extremely low volatility and will stick on the surface when colliding with the particle unless the true mass accommodation coefficient is less than 1. Condensed phase diffusion limitations would cause a substantial activity gradient within the particle, but if the gas-phase activity (the saturation ratio) is » 1, no condensed-phase activity gradients can significantly influence the microphysics (since the condensed-phase activity is the mole or volume fraction depending on the thermodynamic formulation, for all but very small particles < 10 nm or so with significant curvature). Our conclusion here is that the condensable PD products include a large fraction of ELVOCs, which is also strongly indicated by the new-particle formation experiments at CLOUD.

We have looked and looked and looked for indications of substantial diffusion limitations for SVOC mass transfer between SOA particles, and thus far this has been a rare occurrence. From the literature (Saleh et al., 2013), members of our research team found the accommodation coefficients of alpha-pinene SOA to be >~ 0.2. Other members of our team have explored interactions of suspended SOA populations using isotopically labeled precursors and single-particle measurements (Robinson et al, J Phys Chem, 2013; P. Ye et al., J Phys Chem 2014; Q. Ye et al PNAS 2016; Q Ye et al., Chem, 2018). In no case, for experiments spanning the full range of RH, have we found evidence for substantial delays to vapor exchange between particle populations involving SOA formed from alpha-pinene. While we have not directly studied PD products using this method, we regard the alpha-pinene experiments as a useful analogue. For this reason, we treated two limiting cases, alpha = 0.1 and 1.

Minor: 1. Line 211: Specify how long it takes between the chemical injection into the chamber and the measurement of their concentrations by PTRMS/GCMS. What is the chamber mixing timescale?

ANSWERS: The injection time was 15 mins. Tenex tube samples were collected at 15 mins after the injections were completed. PTRMS was sampling all the time. The chamber mixing time is 5-10 mins.

2. Line 252: Please show evidence for the 'rapid vapor-wall equilibrium' observed in the experiments.

ANSWERS: We observed the rapid change of the SVOC concentration change in the gas phase due to the saturation concentration change caused by the temperature vibration in our previous paper (Ye et al., 2016a)

3. Line 295: Again, specify the time duration between chemical injection and the onset of chamber dilution.

ANSWERS: It was around 1 hour

---

## Author Comment (AC2) · 14 Mar 2018

Reviewer 2:

General comments:

1. What 8 m3 chamber has a surface area of 12 m2 (line 105)? This is off by a factor of probably about 2. The smallest surface area to volume ratio is that of a sphere, and a sphere with a volume of 8 m3 would have a surface area of 19.3 m2. Likely, any chamber with this volume would have an even larger surface area (and certainly much

larger than 12 m2). Related to this, what is the source of the estimate of 10 g in line 107?

ANSWERS: 12 m2 is a typo. It should be 24. The chamber is a cubic shape. We used 0.8 g/cm3 as the density of the Teflon to calculate the $1\mu$m thick Teflon layer mass and got 10 g.

2. How are the data points in Fig. 1 obtained, since in Fig. 2 there is a slightly decreasing trend when the concentrations reach "quasi-steady-state"?

ANSWERS: We averaged the concentrations from the time when the gas concentration got stable to right before the next injection.

3. How do you perform the stepwise injection of the compounds in Fig. 2, i.e. at each injection step does the volume of the chamber change because of constant sampling? Also, you mention the longer evaporation time of the less volatile compound: can you give an estimated timescale?

ANSWERS: We put the mixture the compounds in a flash vaporizer consisting of a stainless steel tip with a machined trough for compounds containing a resistive heating element, all inserted well into the chamber at the end of a stainless steel tube through which we passed purified, heated air. We used the purified air flow to transfer the vapors into the chamber while heating the mixture. The total sampling rate from the chamber was around 5L/min. We used 15L/min air flow to inject the organic mixture for 15 mins. It was around 40 mins between each injection. So the injection and sampling flow were almost balanced. The change of the chamber volume is very small. In this study, the evaporation time of pinanediol was around 10 minutes. We used a low heating output to avoid the thermal decomposition of pinanediol.

4. In the heating experiment (Fig. 3), how much PD do you inject into the chamber at 13oC in order to get 866 $\mu$g m-3? Have you tried to increase temperature to just 22oC to see if you can get a similar portion of bulk concentration of PD with the ones in Fig.

2? In other words, how can you verify the possibility of pure condensation of PD on the wall or other lines at such a lower temperature? Otherwise, one would think the vapor-wall interaction mechanism is different in heating and dilution experiments.

ANSWERS: We put 20mg pinanediol in the chamber. We tried a series of different amounts of pinanediol. 866 $\mu$g/m3 was in the middle of the gas phase concentrations we measured. We regarded pure condensation of PD as unlikely since the PD was not saturated in the gas phase. However, it is not obvious at all that this would produce a different result. For "pure condensation" the gas-phase (and condensed-phase) activities would be 1 – the system would be saturated. Consequently, there would be a condensed-phase reservoir with an equilibrium vapor pressure of the PD saturation vapor pressure in the chamber or the lines; this in turn would lead to a significant return flux when the system was dis-equilibrated by dilution. The only substantial difference would be that we would not have been able to add more PD to the gas phase, because it would have been saturated. That is directly contradicted by the data in Figures 1 and 2.

5. In the dilution experiment, you show that PD-wall partition is irreversible above 22% of the initial value, which may be true if the oxidation rate of PD is similar to the dilution rate. So how do you simulate the photo-oxidation of PD? What are the actual values of jHONO and OH level in the chamber? What is the oxidation mechanism used in the simulation: parallel or in series?

ANSWERS: The simulation here was purely experimental. The removal of PD by dilution directly simulates removal of PD by oxidation; there should be no difference to the wall-vapor equilibration because the remaining PD molecules will not "know" how their missing comrades came to vanish - whether down a drain or via oxidation. From the dilution experiment, we found the PD started to release from the chamber wall only after the PD concentration reached 2 $\mu$g/m3. We limited our analysis to the first 1.5 e-folding lifetimes in PD oxidation (we only use the data where the PD concentration is above 8 $\mu$g/m3, 22% of its initial value). For the 2D-VBS simulations we used the constrained (measured) PD removal rate to drive formation of VBS products, again without direct numerical simulation of the gas-phase chemistry.

We injected PD and HONO into the chamber and turned on the UV lights to initiate the oxidation of PD with OH radicals. The OH concentration in these experiments was around $2.4 \times 107$ molecules/cm3 for the first hour, then dropped to around $5 \times 106$ molecules/cm3 afterwards.

6. If the conclusion in lines 318-320 is correct, why does Fig. 1 not have a y-intercept of 0? Also, how are you accounting for the additional loss you saw in the experiment for Fig. 4?

ANSWERS: The y-intercept is a little bit away from 0 may be due to the large uncertainty of the measurement when PD concentration was low. The decrease of PD was very slow, the loss rate is around 0.05/h. This gave a very small uncertainty when calculating the mass yield. Consequently, we just used the PTR measurement to do the calculations.

7. Around line 372, you are assuming that the condensation sink does not change as more vapor deposits throughout the experiment. How do you justify this assumption, particularly for the boundary layer? The mass transport through the boundary later is changing throughout the experiment, so the condensation sink of deposited particles also changes.

ANSWERS: We do not assume that the suspended condensation sink is a constant – we measure the suspended particle surface area, correct it for near-surface diffusion (i.e. Fuchs and Sutugen) and calculate the collision frequency of vapors with that suspended surface area. When alpha=1 this is the condensation sink, when alpha < 1 it the condensation sink is slightly larger than alpha x collision frequency (in the transition regime). For the chamber walls, we assume that the condensation sink to the walls is completely limited by diffusion to the chamber walls and that uptake of vapors is quasi-irreversible. McMurry and Grosjean showed decades ago that this

will be true so long as the accommodation coefficient of vapors to the walls is larger than roughly 1e-4, and in vapor wall loss experiments we have found no evidence that accommodation is delayed; consequently, vapor transfer to the chamber walls is rate limited by gas-phase diffusion in the quasi laminar boundary layer. Members of our team described this in Trump et al, Aerosol Science and Technology, 2016).

8. Can you clarify the necessity of the correction for delayed condensation? In the caption of Fig. 8, you attribute the delayed condensation to the diffusion time of vapor molecules to the surface of the particles or the wall. Do you mean the gas-phase production rate is too fast compared with the timescale to reach gas-particle-wall equilibrium, so that the instantaneous equilibrium assumption cannot be used at the initial stage?

ANSWERS: The delayed condensation will mostly affect the observed SOA mass in the early stage of the experiments, likely the first 20 mins. During this period, the equilibrium may not be obtained instantaneously.

9. Since you are comparing your experiment to a nucleation experiment in CLOUD (lines 513-520), you should justify your assumption that you used enough seed to suppress nucleation when discussing particle number concentration (line 351).

ANSWERS: This is not an assumption - we measured the suspended number concentration and no new particles appeared. We focused on the chemical compositions observed in this study to the CLOUD experiments. Because we did not observe nucleation in these experiments, the seed particles evidently provided enough surface to prevent the nucleating ELVOCs from building a supersaturation sufficient for nucleation. Members of our team modeled this for the alpha-pinene SOA case, comparing SOA production with CLOUD nucleation, in Chuang et al, ACP, 2017. However, for PD in CLOUD, the nucleation involves sulfuric acid vapor and so we cannot directly compare the nucleation results (we do not know when nucleation "should" or "should not" have occurred in our experiments given the product formation rate, suspended condensation

sink, and consequent steady-state supersaturations of nucleating species).

10. How do you distinguish "overall SOA yield" and "instantaneous SOA yield"? It looks like Fig. 9 and Fig. 10 are plots of temporal profile of overall SOA yield.

ANSWERS: The "overall SOA yield" in the manuscript means all SOA yields we observed at different PD initial concentrations. We removed the term "overall" in the revised manuscript. The "instantaneous SOA yield" is the overall SOA yield.

Specific comments:

Line 93: Remove the symbol "‡" in the citation. Lines 91-94 repeat what is more succinctly said in line 89. Line 163: I believe the unit is m3 not m-3. ANSWERS: We changed those in the manuscript. Line 178: What type of neutralizer did you use?

ANSWERS: It is Po-210

Lines 199, 265, 269, 271, 306: There should not be a space before °C. Lines 213 and 220: The period should go after "Fig" not after the number, as is done in the rest of the paper.

ANSWERS: We changed those in the manuscript.

Line 218: Why does it look like the y-intercepts for oxy pinocamphone and PD are not 0?

ANSWERS: The y-intercept is a little bit away from 0 may be due to the large uncertainty of the measurement when PD concentration was low.

Line 228/Fig. 2: The overshoot time for 2-Nonanone appears to be a lot closer to 10 minutes than to 1 min, especially for the data a little after 2 hours.

ANSWERS: We only counted the first peak as the overshoot time in the original manuscript. We changed it to "5 to 10 mins"

Lines 265 and 278-280: These sentences repeat each other but, in line 265, you say

"factor of 10 to 30" and in lines 279-280 you just say "30-fold increase." What happened to the range in the second sentence?

ANSWERS: It should be "factor of 30" in line 265. We changed the wording in the revised manuscript.

Line 283: The PD should be "absorbed into" or "absorbed by" the Teflon walls, not "absorbed in" them.

ANSWERS: We changed it to "absorbed into"

Line 307: Does the ratio decrease before dilution when the concentration is held constant? Otherwise, diffusion into the bulk Teflon does not make sense.

ANSWERS: The ratio also decreased at a similar rate before dilution.

Line 308: This is the wrong Zhang 2015 reference.

ANSWERS: We put in the right reference.

Line 310: There should be a space after "5.5" before "h," as is done in the rest of the paper.

ANSWERS: We changed those in the manuscript.

Line 317: Did you try slowing the rate of dilution even more to see if there was an effect?

ANSWERS: We didn't try a slower dilution rate.

Line 339: "as same as" should be "the same as" or something of that sort. Line 352: The font is bold.

ANSWERS: We changed those in the manuscript.

Line 352: How do you verify ignoring other dependencies? E.g. the dependence of the wall loss rate on the diameter of the particle.

ANSWERS: The wall loss of particles also depends on the particle size. We added "without considering the size dependence particle wall loss and other effects"

Lines 384 and 386: These lines have odd spaces/indentations. Line 437: Inconsistent spacing after the equals sign. Line 474: Be consistent between "oxy-pinocamphone" and "oxy pinocamphone." Line 466-476: It is better to represent the chemical mechanism in a scheme. Line 517: OSC needs a line above it instead of an accent mark.

ANSWERS: We changed those in the manuscript.

Line 535: Where in the supplemental material is this provided?

ANSWERS: It should be "in the following section".

Line 536: You should probably mention this is for $\alpha$=1 and give the justification for choosing this value of $\alpha$ that you give in the figure captions. Lines 561, 880, and 904: "Teflon," "summary," and "simulation" are misspelled. Lines S26-S28: It is unclear when you switch to an explanation of method 3.

ANSWERS: We changed those in the manuscript.

Figure 3: Why is there a bump/overshoot in the Pinanediol concentration around 0.1 hours?

ANSWERS: This is probably due to a combination of chamber mixing and the fact that the heating is delivered directly through the walls - it is not unreasonable to expect a surge of material off of the walls during the initial heating shock. However, this is total speculation.

Figures 4 and S1: Why not make these A and B parts of a figure, so that they can be more directly compared?

ANSWERS: This is a good suggestion - we have combined the figures in the revised manuscript.

Figure 5: The SMPS used in this experiment cannot detect nano-particles, so the last sentence about nucleation may not stand.

ANSWERS: We observe growth rates of the accumulation mode (seed) particles and this constrains the growth rates of nucleated particles as well (they will in general be significantly larger). During the active SOA formation period of these experiments the SOA growth rates exceeded 100 nm/h, so any nucleated particles would have grown into our SMPS detection range in 6 min or less, with a very high survival probability. While it is possible that alien nano-spacecraft where zapping the nucleated particles out of the bag before they grew into our detection window, we regard this as sufficiently unlikely to exclude if from our analysis.

Figure 7: Use another color or background for the case $\alpha$ = 0.1. Figure 7: The solid red versus thickly shaded red are very difficult to distinguish, even when viewed in color. Figure 8: Since you already use red in the figure, it may make more sense to replace the red dashed line with another color.

ANSWERS: We recolored Fig. 7 and 8.

Figure 11: This figure is missing a legend.

ANSWERS: We added a legend.

Figure 12: Missing colorbar for contour lines.

ANSWERS: The contour lines are not colored - they are in the figure for a qualitative representation of the 2D product distribution. The quantitative representation is the sum over O:C (the 1D representation) shown in the lower panel.

Figure 13: I suggest you change "Bulk suspended" to "Particle suspended" in the legend. Figures S2 and S3: Cn is never defined. Also, in S2, the labels on the blue arrows are sufficiently far away from these arrows to be somewhat confusing. ANSWERS: We changed those in the manuscript.